# Semi-Supervised Regression with Heteroscedastic Pseudo-Labels

Xueqing Sun[1], Renzhen Wang[1]*, Quanziang Wang[1], Yichen Wu[2,3], Xixi Jia[4], Deyu Meng[1,5]

[1] Xi'an Jiaotong University [2] City University of Hong Kong [3] Harvard University
[4] Xidian University [5] Pazhou Laboratory (Huangpu)
xqsun@stu.xjtu.edu.cn, {rzwang,dymeng}@xjtu.edu.cn
{quanziangwang, wuyichen.am97, hsijiaxidian}@gmail.com

## Abstract

Pseudo-labeling is a commonly used paradigm in semi-supervised learning, yet its application to semi-supervised regression (SSR) remains relatively under-explored. Unlike classification, where pseudo-labels are discrete and confidence-based filtering is effective, SSR involves continuous outputs with heteroscedastic noise, making it challenging to assess pseudo-label reliability. As a result, naive pseudo-labeling can lead to error accumulation and overfitting to incorrect labels. To address this, we propose an uncertainty-aware pseudo-labeling framework that dynamically adjusts pseudo-label influence from a bi-level optimization perspective. By jointly minimizing empirical risk over all data and optimizing uncertainty estimates to enhance generalization on labeled data, our method effectively mitigates the impact of unreliable pseudo-labels. We provide theoretical insights and extensive experiments to validate our approach across various benchmark SSR datasets, and the results demonstrate superior robustness and performance compared to existing methods. Our code is available at https://github.com/sxq/Heteroscedastic-Pseudo-Labels.

## 1 Introduction

Deep learning has achieved superior performance on various scenarios, particularly when large amounts of labeled data are available [28]. However, acquiring large-scale datasets with accurate labels is often costly or impractical in many real-world scenarios, such as medical imaging where labeled data is typically scarce and expensive or video analysis that requires labor-intensive frame-by-frame annotations [10, 51]. To address this challenge, Semi-Supervised Learning (SSL) has emerged as a potent paradigm that leverages vast amounts of unlabeled data to enhance learning performance, thereby reducing the dependency on expensive labeled annotations [44, 52].

In classification tasks, SSL techniques such as pseudo-labeling [29, 41, 50, 57] and consistency regularization [40, 43, 2, 1, 49] have been widely adopted, achieving impressive results across various domains. However, semi-supervised regression (SSR) presents unique challenges distinct from its classification counterpart. Unlike classification, where pseudo-labels are discrete and can be sharpened to encourage high-confidence predictions, regression inherently deals with continuous outputs, making it difficult to establish a reliable pseudo-labeling mechanism [18]. Furthermore, the lack of well-defined decision boundaries in regression complicates uncertainty quantification, increasing the risk of propagating incorrect pseudo-labels.

Due to these challenges, the pseudo-labeling methods commonly employed in semi-supervised classification are difficult to directly extend to semi-supervised regression due to the continuous

---

*Corresponding author.

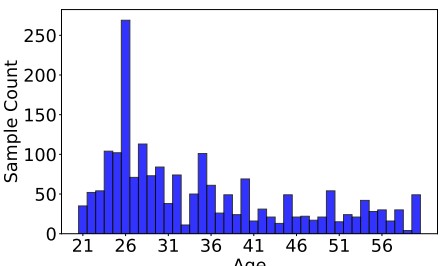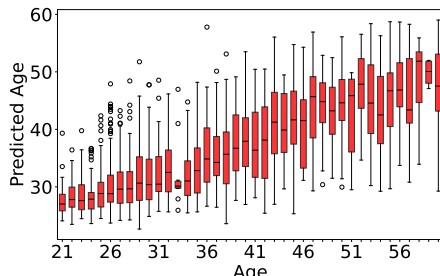

Figure 1: Experiment of UCVME [10] on UTKFace [58] with 10% labeled data. **Left:** Histogram of true labels for the unlabeled data. **Right:** Box-plot of pseudo-labels generated by UCVME.

nature of regression outputs. Consequently, existing SSR approaches focus primarily on leveraging consistency regularization to improve the utilization of unlabeled data. For example, TNNR [47] introduces a loop consistency that enforces consistency on the prediction differences between pairs of inputs. UCVME [10] proposes a novel uncertainty consistency loss for co-trained models to better account for model uncertainty. RankUp [18] reformulates the regression task as a ranking problem and imposes consistency of the rank between sample pairs.

These methods aim to enforce smoothness between predictions for labeled and unlabeled data, ensuring that the model learns a coherent mapping from input features to continuous outputs. However, the reliance on consistency regularization alone does not fully address the difficulties in handling uncertainty and the risk of overfitting to potentially incorrect pseudo-labels. To investigate this, we visualize the distribution of estimated pseudo-labels predicted by UCVME on the UTKFace dataset [58] with 10% labeled data. As shown in Fig. 1, despite the application of uncertainty consistency constraints, UCVME shows significant variance in pseudo-labels (right subfigure), even for age groups with large sample sizes (left subfigure). To address the aforementioned challenge, this paper introduces an uncertainty-aware pseudo-labeling method that dynamically adjusts the influence of pseudo-labeled samples based on calibrated uncertainty.

Our core motivation is that the uncertainty or noise in pseudo-labels varies with input features, leading to heteroscedasticity. Therefore, *it is crucial to develop an effective strategy for assigning appropriate uncertainty values to pseudo-labels, reflecting their varying degrees of error during training.* To achieve this, we propose a bi-level learning framework that not only minimizes empirical risk over both labeled and unlabeled data but also explicitly optimizes uncertainty estimates to improve generalization on labeled data during training. We theoretically and experimentally demonstrate that our method significantly enhances pseudo-label robustness and overall performance across various benchmark SSR tasks. In summary, our main contributions are: **(1)** We propose a bi-level SSR framework that explicitly optimizes the uncertainty of pseudo-labeled samples to improve the regression model's generalization when learning from potentially incorrect pseudo-labels. **(2)** We offer a theoretical analysis of the proposed method through gradient alignment, providing insights into why it enables the regression model to effectively mitigate the risk of overfitting to incorrect pseudo-labels. **(3)** We empirically validate our approach on various benchmark datasets and SSR settings, demonstrating its ability to improve both performance and robustness in SSR tasks.

## 2 Related work

### 2.1 Semi-Supervised Learning

Semi-Supervised Learning (SSL) [44, 52] trains models with both labeled and unlabeled data, which significantly reduces the cost of data annotation in real-world applications. SSL methods have been applied on wild scenarios, such as image classification [25, 41, 2], segmentation [59, 6, 16], and other tasks [54, 51, 33]. Current works on SSL can be categorized as follows: 1) *Pseudo-labeling methods*: These works [29, 41, 50, 57, 46, 17] train the model on both labeled data and unlabeled data with pseudo-labels, which are generated by the model itself. 2) *Consistency-based methods*: These models [40, 43, 2, 1, 49] apply prediction invariance loss on unlabeled data across perturbations. 3) *Generative methods*: These methods [42, 11, 4, 30] aim to model the real data distribution from the labeled and unlabeled data and generate new samples.

## 2.2 Semi-Supervised Regression

Semi-Supervised Regression (SSR), has attracted significant research interest in recent years, which includes 1) *Consistency-based methods*: Similar to the aforementioned methods in SSL, these methods designed different constraints to ensure prediction consistency in SSR. On the one hand, methods developed for semi-supervised classification can be readily extended to SSR, such as Mean Teacher [43] and Temporal Ensembling [26]. On the other hand, methods specifically designed for SSR have emerged. For example, TNNR [47] enforces loop consistency on prediction differences, UCVME [10] introduces an uncertainty consistency loss for co-trained models, and RankUp [18] enforces rank consistency between sample pairs. Additionally, CLSS [9] employs contrastive learning to encourage sample features with the same labels to be closer. 2) *Uncertainty-based methods*: In SSR, uncertainty estimation provides a potential way to improve the quality of pseudo-labels for unlabeled data. Specifically, SSDKL [20] proposes to minimize the prediction variance of the posterior regularization, and SimRegMatch [21] employs dropout to compute multiple predictions and use their variance to quantify the uncertainty of pseudo-labels. In a related manner, PIM [7] performs weakly supervised regression by weighting candidate labels according to their incurred losses, giving higher weights to labels with smaller losses. Unlike these methods, we propose an uncertainty-aware pseudo-labeling method from a bi-level optimization perspective in this work.

## 2.3 Uncertainty Estimation

Uncertainty estimation has been explored extensively in fully-supervised learning [27, 22, 19]. From a Bayesian viewpoint, Gal and Ghahramani [13] reinterpret Monte Carlo Dropout as variational inference for epistemic uncertainty estimation, which has been extended to semi-supervised classification methods [36] and semi-supervised segmentation tasks [56, 48] to filter unreliable pseudo-labels. Additionally, some works [55, 31] quantified uncertainty through prediction discrepancies between co-trained models. In the SSR task, UCVME [10], SSDKL [20], and SimRegMatch [21] have employed the uncertainty estimation methodology to improve the accuracy of pseudo-labels. Concretely, UCVME models pseudo-label noise under the heteroscedastic assumption, using a shared encoder with dual heads to predict both the mean and variance of the noise distribution. SSDKL combines a neural network encoder with a Gaussian Process in the latent space, enabling posterior inference to obtain predictive variance for unlabeled samples. SimRegMatch estimates the uncertainty of pseudo-labels by performing multiple forward passes with dropout, and selects lower uncertainty pseudo-labels through a thresholding strategy. Different from these methods, we propose a bi-level optimization framework and estimate pseudo-label uncertainty guided by labeled samples.

# 3 Method

## 3.1 Notations and Problem Setups

Semi-supervised regression (SSR) involves a labeled dataset $\mathcal{D}_l = \{(x_i^l, y_i^l)\}_{i=1}^N$ and an unlabeled dataset $\mathcal{D}_u = \{x_j^u\}_{j=1}^M$, where $x_i^l \in \mathcal{X}$ and $x_j^u \in \mathcal{X}$ represent training examples from the input space $\mathcal{X}$, and $y_i \in \mathbb{R}$ denotes the label associated with the labeled example $x_i^l$. The objective of SSR is to train a model $f_\theta$ that generalizes well on the test dataset.

In SSR, a widely used strategy involves employing pseudo-labeling techniques to augment the training dataset by assigning pseudo-labels to unlabeled data. In this strategy, each unlabeled example is assigned a pseudo-label based on the predictions of the model itself or its variants. The model is then trained on both labeled and unlabeled examples by optimizing an objective function $\mathcal{L} = \mathcal{L}_l + \lambda \mathcal{L}_u$. The loss function consists of two terms: the supervised loss $\mathcal{L}_l$ calculated on the labeled data, and the unsupervised loss $\mathcal{L}_u$ calculated on the pseudo-labeled data. The hyper-parameter $\lambda$ balances the contributions between the supervised and unsupervised losses.

One of the key challenges in pseudo-labeling-based SSR is to effectively leverage pseudo-labeled data while mitigating the negative impact of incorrect pseudo-labels on model training. In this work, we propose a bi-level learning framework, which explicitly learns pseudo-label uncertainty to enhance the reliability and utility of pseudo-labeled data. By modeling the uncertainty associated with pseudo-labels, the method enables more robust training in SSR settings. In the following section, we first

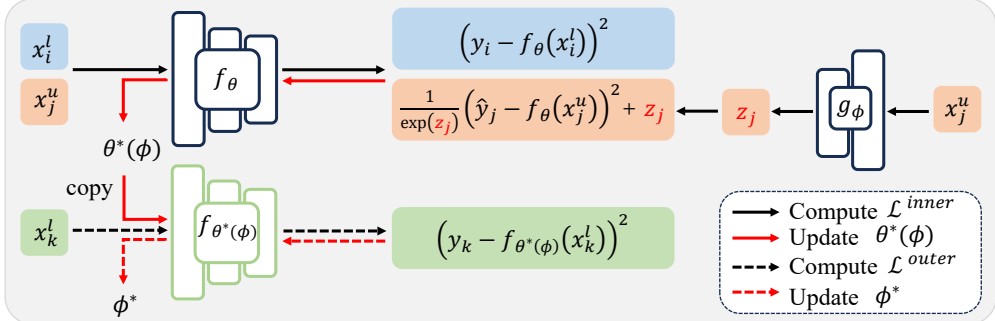

Figure 2: **Method Overview.** The proposed bi-level optimization framework consists of two main steps: (1) Inner-loop update, which updates the regression model using $\mathcal{L}^{inner}$ as defined in eq. (5), where $z_j = \log \sigma_j^2$; (2) Outer-loop update, which updates the uncertainty-learner using $\mathcal{L}^{outer}$ as defined in eq. (6). Note that we assume a batch size of 1 for better visualization.

introduce a probabilistic formulation to model the uncertainty in pseudo-labeling. We then describe how to effectively learn and incorporate pseudo-label uncertainty within our bi-level framework.

### 3.2 Heteroscedastic Pseudo-Labels

The objective of SSR is to learn the true label function $f_{\theta^*}(x)$ using both labeled and unlabeled data. However, when applying pseudo-labeling to SSR, the pseudo-labels assigned to the unlabeled data are usually imperfect. Moreover, the level of uncertainty or noise in these pseudo-labels varies across samples, particularly when the label estimation process involves an external source, such as weak supervision, self-training, or a noisy teacher model.

To model this relationship, we treat the pseudo-labels for each sample as heteroscedastic. Especially, for each unlabeled sample $x_j^u$ and its corresponding pseudo-label $\hat{y}_j$, we assume that the variance of each pseudo-label depends on the input sample itself [3]. This can be formulated as:

$$\hat{y}_j = f_\theta(x_j^u) + \epsilon_j, \quad \epsilon_j \sim \mathcal{N}(0, \sigma_j^2) \tag{1}$$

where $f_\theta(x_j^u)$ is the true label prediction for input $x_j^u$, and $\epsilon_j$ is a noise term whose variance $\sigma_j^2$ varies with the input $x_j^u$. In *maximum likelihood* inference, we can write the negative log-likelihood (NLL) of the observation variable $\hat{y}_j$ as:

$$-\log p(\hat{y}_j \mid x_j^u) \propto \frac{(\hat{y}_j - f_\theta(x_j^u))^2}{\sigma_j^2} + \log(\sigma_j^2). \tag{2}$$

As such, for a batch of unlabeled data $\mathcal{B}_u$, we can define the unsupervised loss as the sum of the negative log-likelihood over all these samples:

$$\mathcal{L}_u = \sum_{x_j^u \in \mathcal{B}_u} \frac{1}{\sigma_j^2}(\hat{y}_j - f_\theta(x_j^u))^2 + \sum_{x_j^u \in \mathcal{B}_u} \log(\sigma_j^2). \tag{3}$$

Notably, if all pseudo-labels are assumed to be homoscedastic and $\sigma$ is fixed at 1, the above loss degrades to the standard mean squared error (MSE) loss, which is commonly employed in various SSR methods.

In eq. (3), when an incorrect pseudo-label $\hat{y}_j$ is assigned to $x_j^u$ during training, the model can mitigate the issue of error accumulation in $f_\theta$ by increasing the uncertainty of the pseudo-label, i.e., adjusting $\sigma_j$ to downweight its contribution to the overall loss. In contrast, the standard MSE loss enforces rigid fitting of incorrect pseudo-labels, thereby exacerbating confirmation bias as training progresses. *Thus, a key challenge is to devise an effective strategy for dynamically assigning appropriate uncertainty values to pseudo-labels with varying degrees of error during training.*

### 3.3 Learning Uncertainty via Bi-level Optimization

To dynamically assign uncertainty values to pseudo-labels, a natural strategy is to introduce an auxiliary network $g_\phi$ parameterized by $\phi$ to produce $\sigma_j$ for each unlabeled sample $x_j^u$, and then

jointly optimize the parameters $\{\theta, \phi\}$ of $f_\theta$ and $g_\phi$ in an end-to-end manner. However, this strategy suffers from a fundamental limitation, as it fails to distinguish between two distinct scenarios:

- Hard but correct samples: The pseudo-label $\hat{y}_j$ is correct, but the model prediction $f_\theta(x_j^u)$ is inaccurate, leading to a large squared error $(\hat{y}_j - f_\theta(x_j^u))^2$.
- Easy but incorrect samples: The pseudo-label $\hat{y}_j$ is incorrect, but the model prediction $f_\theta(x_j^u)$ is already close to the true label, also resulting in a large squared error.

The two scenarios result in a large squared error in the numerator of the first term in eq. (3), leading to a large $\sigma_j^2$ during minimization. However, the increased $\sigma_j^2$ would suppress the learning for hard but correct samples, which is crucial to improve the performance of SSR models.

Ideally, we seek a formulation where $\sigma_j$ selectively suppresses unreliable pseudo-labels while still enforcing learning on difficult but valid samples. To this end, we propose a novel bi-level optimization framework that introduces a supportive model, referred to as the *uncertainty-learner*, to learn the uncertainty associated with pseudo-labels. Specifically, we parameterize the uncertainty-learner as a lightweight network $g_\phi$ that maps each pseudo-labeled sample $x_j^u$ (or certain conditional information derived from the regression model, as elaborated in the next subsection) to its corresponding log-variance. This mapping can be formally expressed as:

$$z_j := \log \sigma_j^2 = g_\phi(x_j^u), \tag{4}$$

where $\phi$ denotes the parameters of $g_\phi$. Note that we predict the log-variance rather than the variance directly, as this approach enhances numerical stability. Directly predicting $\sigma_j^2$ can lead to instability due to potential division by zero in the loss function [23, 22].

Our proposed bi-level optimization framework, which jointly optimizes the regression network $f_\theta$ and uncertainty-learner $g_\phi$, is illustrated in Fig. 2 and can be described in the following two steps:

- **Inner-loop for optimizing $f_\theta$:** The regression network $f_\theta$ is trained on both labeled and pseudo-labeled data by considering the uncertainty of pseudo-labels as follows:

$$\theta^*(\phi) = \arg\min_\theta \mathcal{L}^{inner} := \mathcal{L}_l(\theta) + \lambda \mathcal{L}_u(\theta, \phi), \tag{5}$$

  where $\mathcal{L}_l(\theta) = \sum_{x_i^l \in \mathcal{B}_l} (y_i - f_\theta(x_i^l))^2$ represents the supervised loss computed over a batch of labeled data, and $\mathcal{L}_u(\theta, \phi) = \sum_{x_j^u \in \mathcal{B}_u} \frac{1}{\exp(z_j)}(\hat{y}_j - f_\theta(x_j^u))^2 + \sum_{x_j^u \in \mathcal{B}_u} z_j$ denotes the unsupervised loss computed over a batch of pseudo-labeled data, with $z_j$ obtained from eq. (4). The hyper-parameter $\lambda$ controls the trade-off between $\mathcal{L}_l$ and $\mathcal{L}_u$. In eq. (5), $\phi$ essentially acts as hyper-parameters for $\theta$, and in this step, we just express $\theta^*$ as a function of $\phi$.

- **Outer-loop for optimizing $g_\phi$:** Our ultimate objective is to generate well-calibrated uncertainty estimates that protect the regression model from incorrect pseudo-labels while ensuring good generalization on labeled data. To achieve this, we optimize $\phi$ by continuously tracking the performance of the updated model $f_{\theta^*(\phi)}$ on labeled data, which can be formulated as:

$$\phi^* = \arg\min_\phi \mathcal{L}^{outer} := \sum_{x_k^l \in \hat{\mathcal{B}}_l} (y_k - f_{\theta^*(\phi)}(x_k^l))^2, \tag{6}$$

  where $\hat{\mathcal{B}}_l$ is another batch of labeled data from $\mathcal{D}_l$ that differs from $\mathcal{B}_l$ used in eq. (5), i.e., $\hat{\mathcal{B}}_l \neq \mathcal{B}_l$. This formulation aims to find the optimal hyper-parameter $\phi^*$ such that the regression model, obtained by optimizing eq. (5), should also have good performance over any unbiased and reliable training batch.

In fact, eq. (5) and (6) formulate a bi-level learning framework. In the inner loop eq. (5), the regression model $f_\theta$ is updated using both labeled data and pseudo-labeled data, with the latter calibrated based on the uncertainty estimates generated by $g_\phi$. In the outer loop eq. (6), the parameters of $g_\phi$ are optimized to ensure that the refined model $f_{\theta^*(\phi)}$ achieves more reliable and robust performance.

### 3.4 Practical Implementations

**Uncertainty-learner's architecture.** In practice, the uncertainty-learner $g_\phi$ does not directly map each unlabeled sample to its log-variance $z_j$, as presented in eq. (4). Instead, to improve computational

efficiency, it takes the prediction of the regression model $r_j = f_\theta(x_j^u)$ and the corresponding pseudo-label $\hat{y}_j$ as its input. This mapping can be formally formulated as: $z_j = g_\phi(r_j, \hat{y}_j)$. The architecture of $g_\phi$ is implemented as a multi-layer perceptron (MLP) with one single hidden layer. Despite its simplicity, according to the universal approximation theorem, such a network is theoretically capable of approximating any continuous function defined on a compact set with arbitrary precision [14]. Note that although both $r_j$ and $\hat{y}_j$ are derived from the regression model, they are not strictly identical in this work. This discrepancy arises from the use of different random augmentations during the training process. Inspired by FixMatch [41], we generate the pseudo-label $\hat{y}_j$ from a weakly augmented version of $x_j^u$, while $r_j$ is obtained from a strongly augmented version of the same input.

---

**Algorithm 1** Mini-batch Training Algorithm of the Method

---

1: **Input:** labeled / unlabeled data $\mathcal{D}_l$ / $\mathcal{D}_u$, labeled / unlabeled batch size $n/m$, max iterations $T$
2: **Output:** regression network parameter $\theta^*$
3: Initialize the parameters $\theta^0$ of the regression network and those of the uncertainty-learner $\phi^0$
4: **for** $t = 0$ **to** $T$ **do**
5:     $B_l = \{(x_i^l, y_i^l)\}_{i=1}^n \leftarrow \text{GetBatch}(\mathcal{D}_l, n)$.
6:     $B_u = \{x_j^u\}_{j=1}^m \leftarrow \text{GetBatch}(\mathcal{D}_u, m)$.
7:     $\hat{B}_l = \{(x_k^l, y_k^l)\}_{k=1}^n \leftarrow \text{GetBatch}(\mathcal{D}_l, n)$.
8:     Compute inner loss $\mathcal{L}^{inner}$ by eq. (5).
9:     Update regression network $\theta^{t+1}$ by eq. (7).
10:    Compute outer loss $\mathcal{L}^{outer}$ by eq. (6).
11:    Update uncertainty-learner by $\phi^{t+1}$ by eq. (8).
12: **end for**

---

**Training Algorithm.** The objective function of the regression network $f_\theta$ in eq. (5) and that of the uncertainty-learner $g_\phi$ in eq. (6) formulate a bi-level optimization problem, where the solution of $\theta^*(\phi)$ and $\phi^*$ are nested with each other. Approximately, we update $\theta$ and $\phi$ in a gradient-based method following [32, 12, 35].

(1) **Update $\theta$.** At iteration $t$, we fix the uncertainty-learner parameter $\phi^t$ and conduct one-step gradient descent w.r.t regression network parameter $\theta^t$ in the inner optimization problem eq. (5) as follows:

$$\theta^{t+1}(\phi^t) = \theta^t - \alpha \cdot \nabla_\theta \mathcal{L}^{inner}(\theta^t, \phi^t), \tag{7}$$

where $\alpha > 0$ is the learning rate of the regression network.

(2) **Update $\phi$.** Based on the updated regression network parameter $\theta^{t+1}(\phi)$, we optimize $\phi^t$ in the outer optimization problem eq. (6) by gradient descent:

$$\phi^{t+1} = \phi^t - \beta \cdot \nabla_\phi \mathcal{L}^{outer}(\theta^{t+1}(\phi^t)), \tag{8}$$

where $\beta > 0$ is the learning rate of the uncertainty-learner.

Since $\nabla_\phi \mathcal{L}^{outer}$ in eq. (8) introduces a second-order derivative, it requires unrolling the second-order differentiation of the entire regression network. This process is computationally expensive and inefficient for deep models. To alleviate this computational burden, we introduce an efficient approximation algorithm. Specifically, we assume that $\phi$ is only correlated with the parameters of the regression head (i.e., a single fully connected layer) of $f_\theta$. This allows us to only unroll the second-order derivation of the regression head in eq. (8). Since the regression head contains far fewer parameters than the entire regression network, our proposed algorithm is considerably more efficient than conventional bi-level optimization algorithms [32, 12, 35].

### 3.5 Theoretical Analysis

We herein theoretically analyze the proposed bi-level optimization framework and reveal how our method helps the regression model mitigate the risk of overfitting to incorrect pseudo-labels.

**Theorem 1.** *Let $\nabla_\theta \mathcal{L}^{inner}(\theta, \phi)$ and $\nabla_\theta \mathcal{L}^{outer}(\theta)$ be the gradients of the inner and outer loss w.r.t. $\theta$, respectively. The bi-level optimization in eq. (5) and eq. (6) is equivalent to the following optimization problem:*

$$\min_\phi - \left\langle \nabla_\theta \mathcal{L}^{inner}(\theta, \phi), \nabla_\theta \mathcal{L}^{outer}(\theta) \right\rangle. \tag{9}$$

The proof is presented in Appendix A.1. This theorem demonstrates that optimizing the parameter $\phi$ in our bi-level framework fundamentally aligns the gradients of the inner and outer losses w.r.t. $\theta$. Based on eq. (5) and eq. (6), the gradient-matching formulation in eq. (9) suggests that our method implicitly regularizes the gradients from labeled and unlabeled data under the guidance of the outer data, and the discrepancy between these gradients is modulated by $\phi$ through its influence on the inner loss. Specifically, inaccurate uncertainty estimated by $g_\phi$ would interfere with the inner loss

Table 1: Main results on UTKFace. The performance is reported in the form of 'mean ± std' across six random runs. **Bold** means the best results and our method are shown in gray cells .

| Method | $\gamma = 5\%$ | | $\gamma = 10\%$ | | $\gamma = 20\%$ | |
|---|---|---|---|---|---|---|
| | MAE ↓ | $R^2$ ↑ | MAE ↓ | $R^2$ ↑ | MAE ↓ | $R^2$ ↑ |
| Fully-Supervised | $4.713 \pm 0.039$ | $0.652 \pm 0.006$ | $4.713 \pm 0.039$ | $0.652 \pm 0.006$ | $4.713 \pm 0.039$ | $0.652 \pm 0.006$ |
| Supervised | $6.135 \pm 0.046$ | $0.454 \pm 0.008$ | $5.618 \pm 0.026$ | $0.533 \pm 0.005$ | $5.278 \pm 0.048$ | $0.581 \pm 0.005$ |
| Mean Teacher [43] | $5.912 \pm 0.071$ | $0.476 \pm 0.011$ | $5.474 \pm 0.052$ | $0.537 \pm 0.009$ | $5.126 \pm 0.040$ | $0.584 \pm 0.007$ |
| Temporal Ensembling [26] | $5.963 \pm 0.060$ | $0.478 \pm 0.008$ | $5.437 \pm 0.022$ | $0.557 \pm 0.001$ | $5.123 \pm 0.032$ | $0.604 \pm 0.002$ |
| SSDKL [20] | $5.855 \pm 0.080$ | $0.469 \pm 0.010$ | $5.391 \pm 0.040$ | $0.551 \pm 0.007$ | $5.173 \pm 0.039$ | $0.589 \pm 0.007$ |
| TNNR [47] | $5.817 \pm 0.057$ | $0.505 \pm 0.008$ | $5.372 \pm 0.052$ | $0.563 \pm 0.007$ | $5.104 \pm 0.025$ | $0.602 \pm 0.005$ |
| SimRegMatch [21] | $5.794 \pm 0.073$ | $0.512 \pm 0.020$ | $5.338 \pm 0.065$ | $0.584 \pm 0.009$ | $5.164 \pm 0.108$ | $0.612 \pm 0.011$ |
| UCVME [10] | $5.862 \pm 0.036$ | $0.495 \pm 0.006$ | $5.227 \pm 0.045$ | $0.585 \pm 0.006$ | $\mathbf{4.870 \pm 0.038}$ | $\mathbf{0.634 \pm 0.005}$ |
| CLSS [9] | $6.063 \pm 0.145$ | $0.451 \pm 0.018$ | $5.621 \pm 0.077$ | $0.514 \pm 0.013$ | $5.429 \pm 0.078$ | $0.546 \pm 0.011$ |
| RankUp [18] | $5.719 \pm 0.112$ | $0.495 \pm 0.012$ | $5.252 \pm 0.040$ | $0.576 \pm 0.004$ | $4.914 \pm 0.015$ | $0.621 \pm 0.002$ |
| Ours | $\mathbf{5.639 \pm 0.035}$ | $\mathbf{0.523 \pm 0.009}$ | $\mathbf{5.143 \pm 0.075}$ | $\mathbf{0.597 \pm 0.009}$ | $4.929 \pm 0.038$ | $0.624 \pm 0.004$ |

gradient w.r.t. $\theta$, thus increasing the empirical risk in eq. (9). In contrast, when $g_\phi$ estimates accurate uncertainty in the inner loss, it helps the regression model perform gradient decent in an appropriate direction consistent with the outer data, ensuring stable and effective semi-supervised learning.

# 4 Experiments

In this section, we experiment with three benchmarks to evaluate our method. The experimental setups are described in Section 4.1, and the main results under varying label ratios are presented in Section 4.2. Section 4.3 provides an ablation study to assess the contribution of each key component, along with further discussion to gain a deeper understanding of the proposed method.

## 4.1 Experimental Setups

**Datasets.** We evaluate the effectiveness of our algorithm on three benchmark datasets: UTKFace [58], an image-based age estimation dataset; IMDB-WIKI [39], a large-scale dataset for age estimation; and STS-B [5, 45], a benchmark for assessing semantic similarity between sentence pairs. Please refer to Appendix B.1 for more detailed information about these datasets. These datasets are commonly used in supervised and semi-supervised regression tasks [10, 18, 53, 34]. In this work, we conduct experiments with varying label ratios $\gamma$, where 5%, 10%, and 20% of the training data are labeled, and the remaining data are used as unlabeled samples.

**Evaluation Metrics.** For the image datasets UTKFace and IMDB-WIKI, we follow UCVME [10] in using Mean Absolute Error (MAE, ↓) and the coefficient of determination ($R^2$, ↑) for evaluation. For the text dataset STS-B, we adopt Mean Squared Error (MSE, ↓) and $R^2$, following DIR [53]. Note that ↓ and ↑ indicate that lower and higher values are better, respectively. Detailed definitions of these metrics are provided in Appendix B.2. All experiments are conducted six times using fixed random seeds (0 to 5), and we report the mean and standard deviation for each metric.

**Training Details.** We conduct all experiments within a unified codebase to ensure fair comparisons. Specifically, following [10], we use ResNet-50 [15] pretrained on ImageNet as the backbone for the UTKFace and IMDB-WIKI datasets. For the STS-B dataset, we adopt a BiLSTM-based regression model with GloVe word embeddings, in line with [45, 53]. For more details on the implementation details, please refer to Appendix B.3.

**Comparison Methods.** We compare our method with various state-of-the-art SSR methods, which can be roughly divided into two categories. 1) *Consistency-based methods*: These methods leverage consistency constraints in model predictions, including some adapted from classification methods such as Mean Teacher [43] and Temporal Ensembling [26], as well as methods specifically designed for SSR, including TNNR [47], UCVME [10], CLSS [10] and RankUp [18]. 2) *Uncertainty-based methods*: These approaches explicitly model uncertainty in data or predictions, including SSDKL [20] and SimRegMatch [21]. Please refer to Appendix B.4 for more details about these methods. Moreover, we construct two benchmarks to validate the effectiveness of SSR methods: 1) *Supervised*: A lower bound for SSR methods that is trained on the labeled data without incorporating any unlabeled data. 2) *Fully-Supervised*: An upper bound trained on all labeled and unlabeled data while assuming ground truth labels of unlabeled data are known.

Table 2: Main results on IMDB-WIKI. The performance is reported in the form of 'mean $\pm$ std' across six random runs. **Bold** means the best results and our method are shown in  gray cells .

| Method | $\gamma = 5\%$ | | $\gamma = 10\%$ | | $\gamma = 20\%$ | |
|---|---|---|---|---|---|---|
| | MAE $\downarrow$ | $R^2 \uparrow$ | MAE $\downarrow$ | $R^2 \uparrow$ | MAE $\downarrow$ | $R^2 \uparrow$ |
| Fully-Supervised | $7.974 \pm 0.043$ | $0.724 \pm 0.002$ | $7.974 \pm 0.043$ | $0.724 \pm 0.002$ | $7.974 \pm 0.043$ | $0.724 \pm 0.002$ |
| Supervised | $10.172 \pm 0.077$ | $0.610 \pm 0.004$ | $9.248 \pm 0.052$ | $0.657 \pm 0.002$ | $8.647 \pm 0.099$ | $0.690 \pm 0.005$ |
| Mean Teacher [43] | $9.492 \pm 0.051$ | $0.647 \pm 0.002$ | $8.633 \pm 0.093$ | $0.689 \pm 0.002$ | $8.191 \pm 0.066$ | $0.711 \pm 0.002$ |
| Temporal Ensembling [26] | $11.335 \pm 0.114$ | $0.532 \pm 0.007$ | $9.517 \pm 0.064$ | $0.639 \pm 0.004$ | $9.577 \pm 0.126$ | $0.639 \pm 0.007$ |
| SSDKL [20] | $10.116 \pm 0.073$ | $0.611 \pm 0.004$ | $9.488 \pm 0.031$ | $0.641 \pm 0.002$ | $9.056 \pm 0.043$ | $0.656 \pm 0.003$ |
| TNNR [47] | $10.069 \pm 0.088$ | $0.612 \pm 0.005$ | $9.309 \pm 0.052$ | $0.654 \pm 0.003$ | $8.640 \pm 0.033$ | $0.688 \pm 0.001$ |
| SimRegMatch [21] | $9.908 \pm 0.097$ | $0.628 \pm 0.004$ | $9.110 \pm 0.166$ | $0.665 \pm 0.007$ | $8.587 \pm 0.094$ | $0.693 \pm 0.006$ |
| UCVME [10] | $9.730 \pm 0.156$ | $0.633 \pm 0.007$ | $8.920 \pm 0.039$ | $0.673 \pm 0.004$ | $8.309 \pm 0.117$ | $0.698 \pm 0.003$ |
| CLSS [9] | $9.906 \pm 0.058$ | $0.621 \pm 0.007$ | $9.251 \pm 0.107$ | $0.656 \pm 0.006$ | $8.781 \pm 0.070$ | $0.681 \pm 0.003$ |
| RankUp [18] | $10.251 \pm 0.072$ | $0.599 \pm 0.005$ | $8.836 \pm 0.047$ | $0.676 \pm 0.003$ | $8.216 \pm 0.022$ | $0.703 \pm 0.001$ |
| Ours | $\mathbf{9.177 \pm 0.061}$ | $\mathbf{0.664 \pm 0.003}$ | $\mathbf{8.539 \pm 0.065}$ | $\mathbf{0.695 \pm 0.003}$ | $\mathbf{8.166 \pm 0.071}$ | $\mathbf{0.712 \pm 0.002}$ |

Table 3: Main results on STS-B. The performance is reported in the form of 'mean $\pm$ std' across six random runs. **Bold** means the best results and our method are shown in  gray cells .

| Method | $\gamma = 5\%$ | | $\gamma = 10\%$ | | $\gamma = 20\%$ | |
|---|---|---|---|---|---|---|
| | MSE $\downarrow$ | $R^2 \uparrow$ | MSE $\downarrow$ | $R^2 \uparrow$ | MSE $\downarrow$ | $R^2 \uparrow$ |
| Fully-Supervised | $0.986 \pm 0.024$ | $0.533 \pm 0.012$ | $0.986 \pm 0.024$ | $0.533 \pm 0.012$ | $0.986 \pm 0.024$ | $0.533 \pm 0.012$ |
| Supervised | $1.746 \pm 0.016$ | $0.173 \pm 0.007$ | $1.524 \pm 0.010$ | $0.278 \pm 0.005$ | $1.332 \pm 0.012$ | $0.369 \pm 0.005$ |
| Mean Teacher [43] | $1.751 \pm 0.010$ | $0.170 \pm 0.005$ | $1.607 \pm 0.017$ | $0.240 \pm 0.007$ | $1.409 \pm 0.009$ | $0.332 \pm 0.004$ |
| Temporal Ensembling [26] | $1.683 \pm 0.007$ | $0.203 \pm 0.004$ | $1.554 \pm 0.009$ | $0.264 \pm 0.004$ | $1.374 \pm 0.015$ | $0.349 \pm 0.007$ |
| SSDKL [20] | $1.606 \pm 0.057$ | $0.239 \pm 0.027$ | $1.407 \pm 0.029$ | $0.333 \pm 0.014$ | $\mathbf{1.211 \pm 0.031}$ | $\mathbf{0.426 \pm 0.015}$ |
| TNNR [47] | $1.746 \pm 0.007$ | $0.166 \pm 0.004$ | $1.534 \pm 0.003$ | $0.266 \pm 0.001$ | $1.333 \pm 0.020$ | $0.362 \pm 0.010$ |
| SimRegMatch [21] | $1.714 \pm 0.032$ | $0.188 \pm 0.015$ | $1.559 \pm 0.024$ | $0.261 \pm 0.011$ | $1.437 \pm 0.015$ | $0.319 \pm 0.015$ |
| UCVME [10] | $1.713 \pm 0.009$ | $0.188 \pm 0.004$ | $1.577 \pm 0.016$ | $0.253 \pm 0.008$ | $1.373 \pm 0.020$ | $0.349 \pm 0.009$ |
| CLSS [9] | $1.622 \pm 0.046$ | $0.231 \pm 0.022$ | $1.421 \pm 0.030$ | $0.327 \pm 0.014$ | $1.286 \pm 0.032$ | $0.390 \pm 0.015$ |
| RankUp [18] | $1.844 \pm 0.063$ | $0.126 \pm 0.030$ | $1.454 \pm 0.016$ | $0.311 \pm 0.008$ | $1.279 \pm 0.018$ | $0.394 \pm 0.008$ |
| Ours | $\mathbf{1.540 \pm 0.006}$ | $\mathbf{0.270 \pm 0.003}$ | $\mathbf{1.403 \pm 0.013}$ | $\mathbf{0.335 \pm 0.006}$ | $1.246 \pm 0.015$ | $0.409 \pm 0.008$ |

## 4.2 Main Results

**Results on UTKFace.** Table 1 summarizes the results of our method and other comparison methods on UTKFace dataset with various ratios. It can be observed that: 1) All SSR methods perform better than the benchmark *Supervised*, which is the lower bound trained by labeled samples, demonstrating that incorporating unlabeled data is beneficial for the SSR problem and effectively reduces regression error. For example, under the ratio of 5%, our method reduces the MAE by up to 8.1% and increases $R^2$ by up to 15.2% compared to *Supervised*. 2) Compared to other SSR methods, our method consistently achieves the best or second best results across all ratios. In particular, when the labeled data is scarce ($\gamma = 5\%$), our method outperforms the second best RankUp [18] by 1.4% for MAE and 5.7% for $R^2$. 3) As the proportion of labeled data increases, the performance gains of our method are slightly lower than those of UCVME and RankUp. We attribute this to stronger supervision from labeled data, which likely improves the quality and stability of pseudo-labels. Such results further demonstrate that the well-calibrated uncertainty generated by our method plays an essential role in improving SSR performance, especially in scenarios where the labeled data is scarce.

**Results on IMDB-WIKI.** As shown in Table 2, our method consistently outperforms all comparison SSR methods significantly. For instance, under $\gamma = 5\%$, compared to the second best method Mean Teacher, our method reduces MAE by about 3.3% and increases $R^2$ by 2.6%, which achieves new state-of-the-art results. Note that the performance of our method with 20% labeled data is closer to *Fully-Supervised*, served as an upper bound in the SSR problem. These results further validate the strength of our method. Furthermore, compared with other uncertainty estimation SSR methods like UCVME, our method exhibits significant improvement, demonstrating the effectiveness of our predicted uncertainty by $g_\phi$ in our bi-level optimization framework.

**Results on STS-B.** The results are listed in Table 3. It can be observed that our method achieves the best or second-best performance across all ratios. On the one hand, if the labeled data is limited, such as $\gamma = 5\%$ or 10%, our method improves all these SSR methods on both MSE and $R^2$ with a large

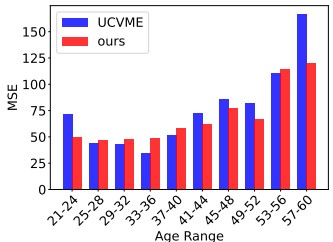

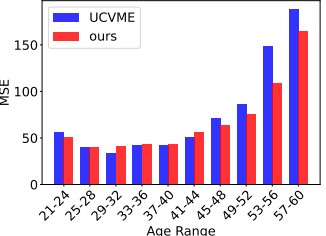

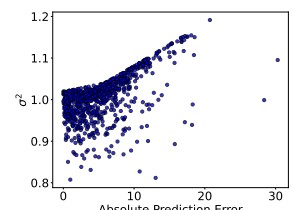

(a) Predictions on unlabeled data    (b) Predictions on test data

Figure 3: Subgroup performance comparison between ours and the second-best (UCVME) on UTKFace with $\gamma = 5\%$.

Figure 4: Correlation analysis between estimated uncertainty $\sigma^2$ and absolute prediction error on IMDB-WIKI with $\gamma = 10\%$ under age 40.

Table 4: Ablation study results on the IMDB-WIKI dataset with label ratios of 5% and 10%. **BL**, **UL**, and **BLO** refer to *Baseline*, Uncertainty-learner and Bi-level Optimization, respectively.

| Components | | | $\gamma = 5\%$ | | $\gamma = 10\%$ | |
|---|---|---|---|---|---|---|
| BL | UL | BLO | MAE $\downarrow$ | $R^2 \uparrow$ | MAE $\downarrow$ | $R^2 \uparrow$ |
| ✓ | ✗ | ✗ | 9.512 | 0.651 | 8.864 | 0.683 |
| ✓ | ✓ | ✗ | 9.914 | 0.630 | 9.562 | 0.651 |
| ✓ | ✓ | ✓ | **9.177** | **0.664** | **8.539** | **0.695** |

Table 5: Computational cost (average training time per iteration) and memory complexity on UTKFace dataset with $\gamma = 10\%$.

| Method | Time (ms) | GPU (MB) |
|---|---|---|
| Baseline | 83.0 | 5099 |
| SimRegMatch[21] | 548.1 | 7419 |
| UCVME [10] | 257.2 | 10057 |
| RankUp[18] | 108.6 | 7433 |
| Ours | 91.6 | 5116 |

margin. For example, when we have only $5\%$ labeled data, MSE of our method is lower than the second-best method SSDKL by 4.1% and $R^2$ is higher by 13.0%. These results demonstrate that our method does not rely on too much labeled data and performs better under such extreme scenarios with scarce labeled samples. On the other hand, our method achieves comparable results with SSDKL when we have more labeled data, *i.e.*, $\gamma = 20\%$. This suggests that our method is not only effective in data-scarce scenarios but also highly adaptable to different levels of labeled data availability.

### 4.3 Discussion and Ablation Study

**Performance analysis across subgroups.** As aforementioned, our method consistently achieves sound performance across three different datasets with various ratios. To further validate the effectiveness of our method, in Fig. 3, we compare the MSE between the ground truth labels and pseudo-labels estimated by UCVME and our method across all ages on UTKFace. We can see that our method can globally produce more accurate pseudo-labels than UCVME across different subgroups, especially on the elder age ranges whose samples are significantly fewer than those of other age ranges, as shown in Fig. 1. These results further demonstrate that our method can consistently improve the accuracy across different subgroups.

**How our algorithm adjusts uncertainty.** To answer this question, we visualize the correlation between the uncertainty estimated by the proposed uncertainty-learner $g_\phi$ and the prediction error of the backbone regression model $f_\theta$. Specifically, for all unlabeled training samples of IMDB-WIKI at age 40, we visualize their corresponding estimated uncertainties $\sigma^2$ and absolute prediction errors in Fig. 4. It can be observed that the estimated uncertainties increase as the prediction errors increase, indicating that the uncertainty-learner tends to assign larger uncertainty for the samples with larger prediction errors and actively reduces the contribution of these samples to the regression model training. Furthermore, the correlation between estimated uncertainties and absolute prediction errors is not a simple linearity, suggesting it is essential to use the uncertainty-learner to estimate the unstable uncertainties of different samples. Additional visualization results are provided in Appendix C.1.

**Ablation study.** We conduct an ablation study to analyze the contribution of each component of our method, with the results presented in Table 4. We begin with the *Baseline* (BL) method, which omits the uncertainty-learner (UL) and applies the loss $\mathcal{L}_l + \lambda \mathcal{L}_u$ with a fixed uncertainty $\sigma_j^2 = 1$ for all unlabeled samples. Compared to our method, the absence of uncertainty estimation leads

to poor performance, confirming its effectiveness in SSR. Next, we assess the roles of the UL and bi-level optimization (BLO) in our method. Interestingly, the model with UL alone (*Baseline + UL*) performs worse than the *Baseline*, likely due to joint training of the UL and the regression model, which leads to inaccurate uncertainty estimation and suppress learning from hard but correct samples, as discussed in Section 3.3. In contrast, incorporating the proposed bi-level optimization framework substantially improves performance, suggesting that it enables more accurate uncertainty estimation and, consequently, more effective training of the regression model. This complies with the conclusion of Theorem 1, which theoretically supports the benefit of bi-level optimization in improving uncertainty estimation.

**Computational cost analysis.** We evaluate the computational cost on a single NVIDIA GeForce RTX 4090 for fair comparison. As shown in Table 5, our method incurs only ~17MB additional GPU memory and less than 9ms extra training time per iteration compared to the *Baseline*, as it unrolls gradients only through the linear regression head to update a small number of parameters $\phi$ in the outer-loop optimization. Compared to recent SSR methods, our method achieves superior performance on all datasets with less time and resource consumption. Further analysis of the computational cost of other recent SSR methods is provided in Appendix C.2.

## 5   Conclusion

We propose an uncertainty-aware pseudo-labeling framework for SSR tasks, which addresses the challenge of heteroscedastic noise in pseudo-labels. Different from the existing methods, our approach dynamically calibrates pseudo-label uncertainty from a bi-level optimization perspective. Such learning paradigm ensures reliable uncertainty estimation that improves generalization of the regression model. Theoretical analysis via gradient alignment and empirical results on benchmark SSR tasks demonstrate the effectiveness of our method, significantly enhancing robustness and accuracy over existing approaches.

**Limitations and Broader Impact.** Despite the promising results achieved by our method in semi-supervised regression, particularly in enhancing the robustness of pseudo-labels through heteroscedastic uncertainty modeling, several limitations remain to be addressed. Notably, our approach does not explicitly consider potential systematic biases present in the labeled or unlabeled data, such as demographic imbalances or domain-specific disparities. If left unmitigated, these biases may be implicitly propagated or even amplified through the pseudo-labeling process. Addressing these issues will be one of the key focuses of our future work. Moreover, our current framework is built under the assumption that both labeled and unlabeled samples can be accessed simultaneously during training. Such a requirement may raise privacy concerns in real-world scenarios involving sensitive user data. Incorporating fairness-aware objectives and privacy-preserving mechanisms into future versions of our framework could further enhance its practical applicability and ethical reliability.

## Acknowledgement

We would like to thank all anonymous reviewers for their constructive suggestions for improving this paper. This work was supported in part by the Major Key Project of PCL under Grant PCL2024A06, Tianyuan Fund for Mathematics of the National Natural Science Foundation of China under Grant 12426105, the China NSFC projects under contracts 62306233, 62476214 and 62372359, and the China Postdoctoral Science Foundation (2024M752550).

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

# Supplementary Material

## A Theoretical Analysis

### A.1 Proof of Theorem 1

In this section, we first review the proposed bi-level learning framework to optimize the parameters of the regression model $f_\theta$ and the uncertainty-learner $g_\phi$. The bi-level optimization framework can be formulated as follows:

$$
\phi^* = \arg\min_\phi \mathcal{L}^{outer}\left(\theta^*(\phi)\right)
$$
$$
\text{s.t.} \quad \theta^*(\phi) = \arg\min_\theta \mathcal{L}^{inner}\left(\theta, \phi^*\right),
\tag{10}
$$

where the specific inner and outer losses can be represented as

$$
\mathcal{L}^{outer}\left(\theta(\phi)\right) = \sum_{x_k^l \in \hat{\mathcal{B}}_l} (y_k - f_{\theta^*(\phi)}(x_k^l))^2
$$
$$
\mathcal{L}^{inner}\left(\theta, \phi\right) = \mathcal{L}_l(\theta) + \lambda \mathcal{L}_u(\theta, \phi)
$$
$$
= \sum_{x_i^l \in \mathcal{B}_l} \left(y_i - f_\theta(x_i^l)\right)^2 + \lambda \sum_{x_j^u \in \mathcal{B}_u} \left[\frac{1}{\exp(z_j)}\left(\hat{y}_j - f_\theta(x_j^u)\right)^2 + z_j\right],
\tag{11}
$$

where $z_j = g_\phi(x_j^u)$ denotes the uncertainty estimated by the uncertainty-learner $g_\phi$. To optimize this bi-level optimization framework, we use a nested gradient-optimization-based method to approximately update the regression model parameter $\theta$ and the uncertainty-learner parameter $\phi$. Specifically, in the inner loop, the updating formulation of $\theta$ at iteration $k$ can be expressed as:

$$
\theta^{t+1}(\phi^t) = \theta^t - \alpha \cdot \nabla_\theta \mathcal{L}^{inner}(\theta^t, \phi^t),
\tag{12}
$$

where $\alpha$ is the learning rate of the inner loss.

$$
\phi^{t+1} = \phi^t - \beta \cdot \nabla_\phi \mathcal{L}^{outer}(\theta^{t+1}(\phi^t)),
\tag{13}
$$

where $\beta$ is the learning rate of the outer loss.

For simplicity of notation, we denote $\nabla_\theta \mathcal{L}^{inner}(\theta, \phi)$ and $\nabla_\theta \mathcal{L}^{outer}(\theta)$ be the gradients of the inner and outer loss w.r.t. $\theta$, respectively. Then we provide the proof of Theorem 1 in the following.

$$
\min_\phi - \left\langle \nabla_\theta \mathcal{L}^{inner}(\theta, \phi), \nabla_\theta \mathcal{L}^{outer}(\theta) \right\rangle.
\tag{14}
$$

*Proof.* For the outer optimization problem, we expand this loss function by the first-order Taylor expansion around $\theta^t$ as follows:

$$
\mathcal{L}^{outer}(\theta^{t+1}(\phi^t)) \approx \mathcal{L}^{outer}(\theta^t) + \left(\theta^{t+1}(\phi^t) - \theta^t\right) \nabla_\theta \mathcal{L}^{outer}(\theta^t).
\tag{15}
$$

Substitute eq. (12) into eq. (15), we obtain

$$
\mathcal{L}^{outer}(\theta^{t+1}(\phi^t)) = \mathcal{L}^{outer}(\theta^t) + \left(-\alpha \cdot \nabla_\theta \mathcal{L}^{inner}(\theta^t, \phi^t)\right) \nabla_\theta \mathcal{L}^{outer}(\theta^t).
\tag{16}
$$

Note that the first term on the right side of eq. (16) is irrelevant to the optimization of $\phi^t$, which can be omitted. Thus, minimizing the outer loop of eq. (6) is equivalent to minimizing the second term in eq. (16). That is,

$$
\min_{\phi^t} \mathcal{L}^{outer}\left(\theta^{t+1}(\phi^t)\right) \Leftrightarrow \min_{\phi^t} - \left\langle \nabla_\theta \mathcal{L}^{inner}(\theta^t, \phi^t), \nabla_\theta \mathcal{L}^{outer}(\theta^t) \right\rangle.
\tag{17}
$$

We thus finish the proof. $\qquad\square$

## A.2 Connection to Bayesian Decision Theory

We make an effort to connect our method to Bayesian decision theory [37, 38] to further interpret our bi-level framework. To make the theoretical analysis more tractable, we restate our bi-level learning formulation with a slight simplification. Specifically, we let the predictive variance $\sigma_j^2 = g_\phi(x_j)$, rather than using its logarithm as in the main text. This modification only involves a log-transform and does not alter the essential behavior of our model. The inner-loop optimization becomes:

$$\theta^*(\phi) = \arg\min_\theta \sum_{i=1}^n (y_i - f_\theta(x_i))^2 + \lambda \sum_{j=1}^m [\frac{1}{g_\phi(x_j)}(\hat{y}_j - f_\theta(x_j))^2 + \log g_\phi(x_j)], \quad (18)$$

and the outer-loop optimization is:

$$\phi^* = \arg\min_\phi \sum_{(x_k, y_k) \in \hat{\mathcal{B}}_l} (y_k - f_{\theta^*(\phi)}(x_k))^2, \quad (19)$$

which seeks optimal uncertainty weights $g_\phi(x_j)$ to guide the model $f_{\theta^*(\phi)}$ toward better generalization.

To link our bi-level learning framework to a Bayesian decision problem, we formulate the following decision problem. Concretely, the uncertainty estimator $g_\phi : \mathcal{X} \to \mathbb{R}^+$ defines a randomized (mixed) decision rule by assigning weights to unlabeled samples $x_j \in \mathcal{D}_u$. The model parameter $\theta$ is treated as the latent state, and we define the utility function as:

$$u(g_\phi, \theta) = \log P(\mathcal{D}_l \cup \mathcal{D}_u | \theta) = \log \prod_{x_i \in \mathcal{D}_l} \mathcal{N}(y_i | f_\theta(x_i), \sigma^2) \prod_{x_j \in \mathcal{D}_u} \mathcal{N}(\hat{y}_j | f_\theta(x_j), g_\phi(x_j)). \quad (20)$$

where $\hat{y}_j$ denotes the pseudo-label for $x_j$. Under this formulation, the Bayes-optimal uncertainty function $g_\phi^*$ is the one that maximizes the expected utility w.r.t the posterior distribution over $\theta$, i.e., $\phi^* = \arg\max_\phi \mathbb{E}_{\theta | \mathcal{D}_l}[u(g_\phi, \theta)]$.

We now show an intuition connection between our bi-level optimization and this Bayesian formulation. Specifically, the inner-loop optimization is equivalent to minimizing the negative log-likelihood of the data under the generative model defined earlier. Therefore, $\theta^*(\phi)$ can be interpreted as a MAP estimate of model parameters $\theta$, conditioned on the fixed uncertainty function $g_\phi$. The outer-loop optimization aims to find the optimal uncertainty weights that lead to a model $f_{\theta^*(\phi)}$ with best generalization performance. While this is not a direct maximization of expected utility $\mathbb{E}_\theta[u(g_\phi, \theta)]$, we argue the following: (i) If the posterior $p(\theta | \mathcal{D}_l)$ is sharply peaked (posterior concentration), and (ii) if the outer-loop loss over $\hat{\mathcal{B}}_l$ is a reliable proxy for risk under $p(x, y)$, then minimizing the outer-loop loss approximately maximizes the expected utility, and thus $\phi^*$ approximates the Bayes-optimal mixed-action policy.

## A.3 Pseudo-Label Smoothness Property

Our framework implicitly enforces a smoothness property on the generated pseudo-labels, as supported by both theoretical analysis and empirical evidence:

- **Theoretical smoothness guarantee.** The Lipschitz continuity of both the regression model $f_\theta$ and the uncertainty estimator $g_\phi$ ensures that nearby unlabeled inputs yield similar pseudo-labels and uncertainty estimates. Specifically, for any two unlabeled samples $x_1^u$ and $x_2^u$, we have: $|\hat{y}_1 - \hat{y}_2| \le L_f |x_1^u - x_2^u|$ and $|\sigma_1^2 - \sigma_2^2| \le L_g L_f |x_1^u - x_2^u|$, where $L_f$ and $L_g$ are the Lipschitz constants of $f_\theta$ and $g_\phi$, respectively. These constants can be effectively controlled by architectural design. For example, we can apply spectral normalization to $f_\theta$ to limit its Lipschitz constant, and we design $g_\phi$ as a shallow MLP with ReLU activations and small-initialized weights to promote smoothness.

- **Empirical evidence.** To further validate the smoothness of pseudo-labels in practice, we conducted a variance analysis across age groups on the IMDB-WIKI dataset. Specifically, we computed the standard deviation of predicted pseudo-labels among samples that share the same ground-truth age and compared the results with UCVME[10], which explicitly enforces local consistency via consistency loss. As reported in the table 6, our method consistently yields lower or comparable standard deviations than UCVME in most age groups. This suggests that our pseudo-labels are more stable and consistent across similar inputs, thereby indirectly capturing the desired smoothness prior.

Table 6: Performance comparison across age groups.

| Method | Age=10 | 20 | 30 | 40 | 50 | 60 | 70 | 80 | 90 |
|---|---|---|---|---|---|---|---|---|---|
| UCVME [10] | 10.623 | 6.522 | 5.009 | 5.878 | 7.772 | 10.666 | 12.936 | 13.085 | 5.017 |
| Ours | 3.290 | 6.389 | 6.157 | 6.971 | 8.558 | 10.231 | 10.301 | 6.364 | 4.011 |

# B  Detailed Experimental Setup

## B.1  Dataset Details

In this work, we experiment with three benchmark datasets: UTKFace [58], IMDB-WIKI [39], and STS-B [5, 45], which are detailed as follows:

**UTKFace[58].** The dataset is an image age estimation dataset, where the objective is to predict an individual's age based on a facial image. The dataset consists of 23,705 face images with the labels ranging from 1 to 116 years old. Following the protocols used in UCVME [10], this paper considers only samples aged 21 to 60. The resulting modified dataset includes 10,518 training samples, 3,287 testing samples, and 2,629 validation samples.

**IMDB-WIKI [39].** The dataset is a larger-scale age estimation dataset which collected over 523K facial images and their corresponding age. Following [53], we utilize curated datasets consisting of 191.5K images for training and 11.0K images for validation and testing. The ages range from 0 to 186 years old and the number of images per age varies from 1 to 7149.

**STS-B [5, 45].** Semantic Textual Similarity Benchmark (STS-B) dataset, which consists of 7.2K sentence pairs collected from real worlds, such as news, image and video captions, and natural language inference data. The target of each pair is a continuous similarity score ranged from 0 to 5. Following DIR [53], we construct the training set containing 5.2K pairs, and both the balanced validation set and the test set containing 1K pairs each.

## B.2  Evaluation Metrics

To assess the performance of comparison algorithms, we adopt Mean Absolute Error (MAE $\downarrow$) and the coefficient of determination ($R^2 \uparrow$) for image datasets following UCVME [10], while employing Mean Squared Error (MSE $\downarrow$) and $R^2$ metrics for text datasets following DIR [53].

**Mean Absolute Error (MAE $\downarrow$).** MAE measures the average absolute difference between predictions and ground truths, offering a simple and robust metric for evaluating prediction accuracy. *i.e.*,

$$\text{MAE} = \frac{1}{n} \sum_{i=1}^{n} |f_{\theta^*}(x_i) - y_i|, \tag{21}$$

where $f_{\theta^*}(x_i)$ denotes the prediction for the sample $x_i$, and $y_i$ is the corresponding ground truth.

**Coefficient of Determination ($R^2 \uparrow$).** $R^2$ reflects the goodness of fit of the model to the data. *i.e.*,

$$R^2 = 1 - \frac{\sum_{i=1}^{n} (f_{\theta^*}(x_i) - y_i)^2}{\sum_{i=1}^{n} (y_i - \overline{y})^2}, \tag{22}$$

where $\overline{y}$ is the mean of $y_i$.

**Mean Squared Error (MSE $\downarrow$).** MSE measures the average squared difference between predictions and ground truths, emphasizing larger errors due to the squaring operation. *i.e.*,

$$\text{MSE} = \frac{1}{n} \sum_{i}^{n} (f_{\theta^*}(x_i) - y_i)^2. \tag{23}$$

## B.3  Training Details

**UTKFace and IMDB-WIKI.** Following the training protocol in UCVME [10], we utilize ResNet-50 [15] pretrained on the ImageNet dataset as the backbone regression model. The model is trained by Adam optimizer [24] for 30 epochs, with a learning rate of $10^{-4}$ for the feature extractor and $10^{-3}$ for the regression head. Additionally, we conduct random cropping and horizontal flipping as

weak augmentation, and RandAugment [8] as strong augmentation in the data augmentation process of SSL. As for $g_\phi$, it is optimized by Adam optimizer with a learning rate of $10^{-4}$.

**STS-B.** Following [45, 53], we adopt a BiLSTM architecture with GloVe word embeddings as the backbone regression model, which is trained by Adam optimizer for 200 epochs, with a learning rate of $10^{-4}$ for the feature extractor and $10^{-3}$ for the regression head. For data augmentation, no augmentation is applied to the labeled dataset, while synonym replacement and insertion operations are used as strong augmentation for the unlabeled dataset.

### B.4  Comparison Methods

To comprehensively evaluate the proposed method, we compare it with some state-of-the-art semi-supervised regression methods, including: Consistency-based methods: Mean Teacher [43], Temporal Ensembling [26], TNNR [47], UCVME [10], CLSS [10], RankUp [18]; Uncertainty-based methods: SSDKL [20], SimRegMatch [21].

In this section, we provide a brief introduction to comparison algorithms.

- **Mean Teacher** [43] averages model weights to form a target-generating teacher model via exponential moving average (EMA), and computes the consistency loss between the teacher's predictions and the student's outputs.

- **Temporal Ensembling** [26] maintains an exponential moving average of label predictions on each training example, and penalizes predictions that are inconsistent with this target.

- **TNNR** [47] is trained to predict differences between the labels of the input pair and follow the principle that the loop of predicted differences should sum to zero.

- **UCVME** [10] improves training by generating high-quality pseudo-labels and uncertainty estimates for heteroscedastic regression.

- **CLSS** [9] uses the recovered ordinal relationship for contrastive learning on unlabeled samples to allow more data to be used for feature representation learning.

- **RankUp** [18] converts the original regression task into a ranking problem and training it concurrently with the original regression objective. It introduces two components: the Auxiliary Ranking Classifier (ARC) and Regression Distribution Alignment (RDA).

- **SSDKL** [20] is a non-parametric kernel learning methods based on minimizing predictive variance in the posterior regularization framework. It combines the hierarchical representation learning of neural networks with the probabilistic modeling capabilities of Gaussian processes.

- **SimRegMatch** [21] is a semi-supervised regression framework with two primary modules: uncertainty-based filtering and similarity-based pseudo-label calibration. It filters reliable pseudo-labels using uncertainty and refines them by incorporating labeled data through similarity-based weighting.

## C  Additional Experiment Results

### C.1  More Visualization Results

To further analyze the correlation between estimated uncertainty $\sigma^2$ and absolute prediction error, we present additional age-wise visualizations on the IMDB-WIKI dataset. Consistent with observations in Fig. 5, the uncertainty $\sigma^2$ increases alongside the absolute prediction error across age groups, highlighting the role and effectiveness of the uncertainty-learner.

### C.2  Computational Resources

Compared to other SSR methods, our proposed algorithm achieves superior performance while maintaining significantly lower computational and memory costs. Among the algorithms compared in table 5: SimRegMatch [21] estimates pseudo-label uncertainty through Monte Carlo sampling, which requires multiple forward passes, and stores feature vectors of samples to compute the similarity matrix. UCVME [10] relies on a multi-model architecture and aggregates predictions from repeated forward passes to compute uncertainty-based consistency loss. RankUp [18] not only

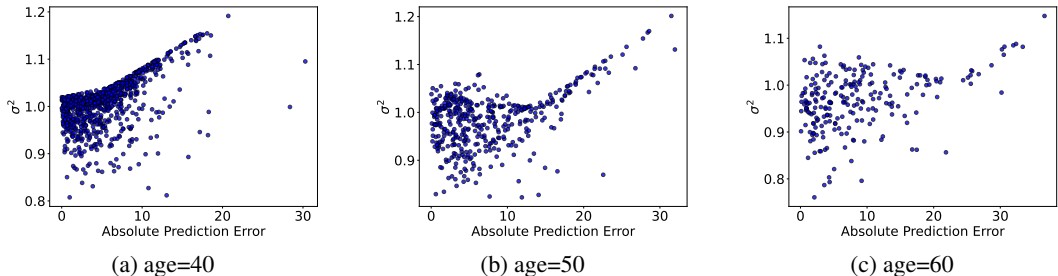

|  (a) age=40 | (b) age=50 | (c) age=60 |

Figure 5: Correlation analysis between estimated uncertainty $\sigma^2$ and absolute prediction error on IMDB-WIKI with $\gamma = 10\%$ under different age.

outputs regression predictions, but also generates ranking scores for each sample, necessitating the computation of all pairwise ranking losses. Furthermore, its RDA module involves frequent lookups from a pseudo-label table during training, which adds additional time overhead. Our method does not require multiple forward passes or the storage of additional intermediate variables during computation. Meanwhile, the proposed uncertainty-learner is a lightweight network; in the optimization process, we assume that the parameters of the uncertainty-learner are only related to the regression head, which significantly reduces computational time and resource consumption.

## C.3 Additional Experimental Comparisons

To further assess the generality and robustness of our method, we additionally include methods from weakly supervised learning (PIM [7]), semi-supervised classification (UPS [36] and CADR [17]).

We provide a brief introduction to these methods below.

- **PIM** [7] computes its loss as a weighted sum of the predictive losses over candidate labels, where labels with smaller losses are assigned higher weights through a softmax-based weighting function.

- **UPS** [36] uses Monte Carlo Dropout to estimate uncertainty via the variance of multiple forward passes, and applies a threshold to identify low-confidence predictions as negative labels, thereby enabling negative learning.

- **CADR** [17] leverages the prior distribution of labeled samples to perform inverse probability weighting (CAP module) and adjusting per-class selection thresholds for unlabeled samples (CAI module).

**Adapting existing methods for SSR.** Several of the compared methods cannot be directly applied to regression tasks, so we propose reasonable adaptations to enable a fair comparison under the SSR setting. Specifically, (1) PIM is originally designed for partial-label regression, where each unlabeled instance is associated with a candidate label set. Since such candidate sets are absent in semi-supervised regression, we construct pseudo-label candidates via Monte Carlo Dropout and compute the PIM loss using a softmax-based weighting over these candidates. (2) UPS relies on defining negative targets in a discrete label space, which is not directly applicable to continuous regression outputs. To adapt UPS fairly, we retain only the uncertainty-based pseudo-label estimation via Monte Carlo Dropout and remove the negative labeling mechanism, yielding the adapted variant UPS_Reg. (3) CADR builds on class-specific confidence modeling, which presupposes discrete label categories. To make it applicable to regression, we discretize the continuous target into age bins and treat each bin as a surrogate class for confidence estimation.

**Results Analysis.** As shown in table 7, our proposed method significantly outperforms all comparison methods across multiple settings. Under the ratio of 20%, our method outperforms the second best method PIM by 3.3% for MAE and 1.1% for $R^2$.

*Comparison with weighting-based methods.* Our uncertainty-aware method and PIM both share the common idea of assigning different loss contributions to samples. However, there is a fundamental difference in how the weights are generated. PIM assigns weights solely based on the prediction

Table 7: Experimental comparison with more methods on the IMDB-WIKI dataset. **Bold** means the best results and our method are shown in gray cells .

| Method | $\gamma = 5\%$ | | $\gamma = 10\%$ | | $\gamma = 20\%$ | |
| --- | --- | --- | --- | --- | --- | --- |
| | MAE ↓ | $R^2$ ↑ | MAE ↓ | $R^2$ ↑ | MAE ↓ | $R^2$ ↑ |
| PIM [7] | $9.345 \pm 0.100$ | $0.659 \pm 0.005$ | $8.691 \pm 0.021$ | $0.691 \pm 0.001$ | $8.441 \pm 0.014$ | $0.704 \pm 0.001$ |
| UPS_Reg [36] | $9.430 \pm 0.087$ | $0.647 \pm 0.006$ | $8.845 \pm 0.055$ | $0.676 \pm 0.003$ | $8.415 \pm 0.002$ | $0.699 \pm 0.001$ |
| CADR [17] | $10.592 \pm 0.091$ | $0.562 \pm 0.009$ | $9.531 \pm 0.064$ | $0.622 \pm 0.007$ | $8.536 \pm 0.056$ | $0.668 \pm 0.008$ |
| Ours | $\mathbf{9.177 \pm 0.061}$ | $\mathbf{0.664 \pm 0.003}$ | $\mathbf{8.539 \pm 0.065}$ | $\mathbf{0.695 \pm 0.003}$ | $\mathbf{8.166 \pm 0.071}$ | $\mathbf{0.712 \pm 0.002}$ |

error of pseudo-labels within each batch, making it prone to reinforcing inaccurate pseudo-labels and amplifying noise. In contrast, our method infers per-sample uncertainty and adaptively adjusts each sample's contribution in a continuous manner, resulting in more reliable weighting.

*Comparison with classification-derived SSL methods.* The inferior performance of UPS and CADR highlights the challenge of directly adapting classification-based methods to regression tasks. Morevoer, compared with UPS_Reg, while both methods incorporate uncertainty, our method uses uncertainty-aware soft weighting within a bi-level optimization framework, yielding more stable and accurate uncertainty estimation.

