# OpenReview forum: "Semi-Supervised Regression with Heteroscedastic Pseudo-Labels"
_NeurIPS.cc/2025/Conference — NeurIPS 2025 poster_

### Official Review · Reviewer_LePr · 2025-06-23

**Clarity:** 2
**Significance:** 2
**Originality:** 3
**Rating:** 4
**Confidence:** 3

**Summary:**

This paper studies pseudo-labeling in the semi-supervised regression (SSR) setting.
In this context, the pseudo-labels are modeled as heteroscedastic noisy observations of the true ones.
As the authors point out, unlike in classification, naive training with noisy pseudo-labels in SSR may lead to poor results due to the continuous nature of the outputs.
To address this, the authors propose a bi-level optimization framework that: (i) estimates the noise level of each pseudo-labeled sample via a neural network $g_\phi$, and (ii) construct an accurate predictor $f_\theta$ using both the labeled and pseudo-labeled samples with appropriate weighting.
They also present numerical experiments demonstrating the advantages of their approach compared to existing methods.

**Questions:**

* Is the noisy-label framework introduced in Equation (1) used in prior works? If so, it would be helpful to mention and cite them.

* Learning with noisy labels is a well-studied area. In particular, in some cases, training directly on noisy samples can still yield good generalization (e.g., linear regression [1]). It would be interesting to better understand whether the proposed scheme achieves improved generalization compared to naive training with noisy labels, even for simple model classes such as linear predictors.


[1] Bartlett, P.L., Long, P.M., Lugosi, G. and Tsigler, A., 2020. Benign overfitting in linear regression.


## Minor comments:
* Line 121: Did you nean *noise level*?
* Algorithm 1, line 7: replace $i$ with $k$
* Experimental setups 4.1: it would be better to define $\gamma$ explicitly in the text
* Line 119: "The objective is to learn the *true* label function $f_\theta (x)$.." . Does it mean that $y \stackrel{\text{a.s.}}{=}f_\theta(x)$?  If so, it may be better to denote the ground truth function by $f*$ (or $f_{\theta^*}$) to avoid confusion with the learned predictor.

**Ethical Concerns:**

["NO or VERY MINOR ethics concerns only"]

**Final Justification:**

The authors addressed my main concerns.

**Limitations:**

Yes

**Quality:**

2

**Strengths And Weaknesses:**

# Strengths:
* Pseudo labeling in SSR setting is very intesrting research area.
* The authors suggest an intersting idea as bi-level optimization approach to deal with the uncertienty of the pseudo labels
* The paper includes extensive experiments.

# Weaknesses:
* Noisy model: The authors modeld the psuedo-labels as noisy observations. However, in practice, pseudo-labels are typically generated by model (either the one trained over the labeled data or an auxiliary weaker model). Therefore, the proposed framework may not capture this case. In particular, when two unlabeled samples $x_1,x_2$ are close, we might expect their pseudo-labels to also be close, a property not directly reflected in the proposed noise model.


* Theoretical results: Although this paper is primarily methodological, the theoretical results are rather limited and do not provide sufficient insight. It would be interesting to understand whether the proposed method reduces the sample complexity in terms of labeled data, or to characterize the noise regimes in which the method is expected to perform well, even for linear regression.

* Uncertainty Learner: A key component of the method is estimating the noise level of each sample by $g_\phi$. However, it only mentioned that $g_\phi$ is an auxiliary network (L. 143).  I encourage the authors to expand this discussion. For example, what are classical approaches for estimating noise levels in noisy labels? What architectural choices are typically used for networks estimating label noise?

---

> ### Author Rebuttal · Authors · 2025-07-30
>
> We sincerely thank you for yur valuable feedback. We hope that our response satisfactorily addresses the issues you raised. Please feel free to let us know if you have any additional concerns or questions.
>
> **W1.** Noisy model: The authors model the pseudo-labels as noisy observations. However, in practice, pseudo-labels are typically generated by model (either the one trained over the labeled data or an auxiliary weaker model). Therefore, the proposed framework may not capture this case. In particular, when two unlabeled samples $x_1$, $x_2$ are close, we might expect their pseudo-labels to also be close, a property not directly reflected in the proposed noise model.
> > We thank the reviewer for raising this insightful concern. While it is true that pseudo-labels are generated by a model rather than directly observed, we respectfully argue that our framework implicitly enforces a smoothness property on the generated pseudo-labels. We support this claim with both theoretical and empirical evidence:
> > + **Theoretical smoothness guarantee.** The Lipschitz continuity of both the regression model $ f_\theta $ and the uncertainty estimator $ g_\phi $ ensures that nearby unlabeled inputs yield similar pseudo-labels and uncertainty estimates. Specifically, for any two unlabeled samples $x_1^u $ and $ x_2^u$, we have: $|\hat y_1 - \hat y_2| \leq L_f \|x_1^u - x_2^u\|$ and $|\sigma_1^2 - \sigma_2^2| \leq L_g L_f \|x_1^u - x_2^u\|, $ where $L_f$ and $L_g$ are the Lipschitz constants of $f_\theta$ and $g_\phi$, respectively. These constants can be effectively controlled by architectural design. For example, we can apply spectral normalization to $f_\theta$ to limit its Lipschitz constant, and we design $g_\phi$ as a shallow MLP with ReLU activations and small-initialized weights to promote smoothness.
> > + **Empirical evidence.** To further validate the smoothness of pseudo-labels in practice, we conducted a variance analysis across age groups on the IMDB_WIKI dataset. Specifically, we computed the standard deviation of predicted pseudo-labels among samples that share the same ground-truth age and compared the results with UCVME [1], which explicitly enforces local consistency via consistency loss. As reported in the table below, our method consistently yields lower or comparable standard deviations than UCVME in most age groups. This suggests that our pseudo-labels are more stable and consistent across similar inputs, thereby indirectly capturing the desired smoothness prior.
> >
> > We hope this analysis clarifies that our framework, despite modeling pseudo-labels as noisy observations, inherently incorporates the smoothness property expected from realistic model-generated pseudo-labels.  We will add this discussion in the revised version.
>
> | Method   | Age=10 | 20 | 30 | 40 | 50 | 60 | 70 | 80 | 90 |
> |-|-|-|-|-|-|-|-|-|-|
> | UCVME [1] | 10.623| 6.522 | 5.009 | 5.878 | 7.772 |10.666 |12.936 |13.085 | 5.017 |
> | Ours  | 3.290 | 6.389 | 6.157 | 6.971 | 8.558 |10.231 |10.301 | 6.364 | 4.011 |
> |
>
> **W2. & Q2.** It would be interesting to understand whether the proposed method reduces the sample complexity in terms of labeled data, or to characterize the noise regimes in which the method is expected to perform well, even for linear regression.
> > We thank the reviewer for raising this important point. While our work is primarily methodological, we agree that offering theoretical insights is valuable. Below, we outline a preliminary theoretical result that helps characterize how our uncertainty-aware pseudo-labeling affects generalization.
> > + **Notations.** We consider the standard regression setting with true risk defined as $ R(f) = \mathbb E_{(x, y)}[\ell(f(x), y)],$ and empirical risk $\widehat R(f) = \widehat R_l(f) + \widehat R_u(f),$ where $\widehat R_l(f) = \frac{1}{N} \sum_{i=1}^{N} \ell(f(x_i^l), y_i^l)$ and $ R_u(f) = \frac{1}{M} \sum_{j=1}^{M} \ell(f(x_j^u), y_j^u).$ Since unlabeled targets $y_j^u$ are inaccessible, we train with pseudo-labels $\hat y_j^u$ by minimizing $\widetilde R_u(f) = \frac{1}{M} \sum_{j=1}^{M} \ell(f(x_j^u), \hat y_j^u).$ Let $\mathcal F$ be the hypothesis space, and $\mathcal R_{N+M}$ be its expected Rademacher complexity over $N+M$ training samples. Let $\hat f = \arg\min_{f \in \mathcal F} \widehat R(f)$ be the empirical risk minimizer, and $f^* = \arg\min_{f \in \mathcal{F}} R(f)$ be the true minimizer. Our inner-loop loss can be denoted as $\mathcal L^{inner}=\frac{1}{N}\sum_{i=1}^{N}\ell(f(x_i), y_i)+\frac{1}{M}\sum_{j=1}^M\frac{1}{\sigma_j^2}\ell(f(x_j),\hat y_j)$.
> >+ **Theoretical Understanding.** Under the assumptions that the loss $\ell(f(x), y)$ is $\mu$-Lipschitz and bounded by $B$, and for unlabeled sample $x_j^u$, that the pseudo-label follows $\hat y_j \sim \mathcal{N}(f(x_j), \sigma_j^2)$ and the model prediction of $f(x_j)$ lies between its true label $y_j$ and pseudo-label $\hat y_j$ as a weighted average, we can show that, for any $\delta>0$, with probability at least $1 - \delta$, it satisfies: $|R(\hat f) - R(f^*)| \leq \frac{1}{M}\sum_{j=1}^{M} \frac{|1 - \sigma_j|}{1 + \sigma_j^2}(y_j - \hat y_j)^2 + 2\mu \mathcal{R}_{N+M} + B \sqrt{\frac{\log(1/\delta)}{2(N+M)}}.$ The first term illustrates that the generalization error is partially governed by the pseudo-label quality and the associated uncertainty $\sigma_j^2$. Specifically, larger uncertainties leads to down-weighting the contribution of high-noise pseudo-labels, while reliable pseudo-labels with lower uncertainty receive higher weight. Hence, learning uncertainty-aware weights provides a theoretically grounded mechanism to mitigate the effect of noisy pseudo-labels and better align the empirical risk with the true target loss.
> >
> > We believe this lays the foundation for deeper investigations into sample complexity and robustness under different noise regimes, which we plan to pursue in future work.
>
> **W3.**  Expand the discussion about uncertainty-learner. For example, what are classical approaches for estimating noise levels in noisy labels? What architectural choices are typically used for networks estimating label noise?
>
> > As suggestion, we will add a more detailed discussion of the noise levels estimation module would as follows.
> > + Estimating sample-wise noise or uncertainty is a well-studied problem. Classical methods can be broadly categorized as follows:
> >    + **Bayesian-inspired methods.** These methods treat outputs as distributions rather than point estimates, often using dropout as a variational Bayesian approximation. Notable works applying dropout to estimate noise or uncertainty include SimRegMatch [4], the method proposed in reference [1], as well as UA-MT [5] and UMCT [6], which use Monte Carlo dropout to quantify pseudo-label uncertainty.
> >    + **Co-train methods.** These methods quantified uncertainty through prediction differences between co-trained models. For example, the method [7] and [8] estimate uncertainty based on prediction discrepancies.
> >    + **Non-parametric kernel methods.** Such methods do not assume a parametric form of the predictive distribution. SSDKL [3] minimizes predictive variance within a posterior regularization framework, combining neural networks for representation learning with Gaussian processes for probabilistic regression.
> >    + **Heteroscedastic regression.** These methods assume that the noise level varies with input samples. For example, UCVME [1] employs a consistency loss under the heteroscedastic assumption to directly learn the noise in data labels through the model.
> > + Other architectural choices for networks estimating label noise include: UCVME [1] adopts a shared encoder with dual heads to predict the mean and variance of the noise distribution, while SSDKL leverages Gaussian Processes to estimate the predictive variance for unlabeled samples.
> >
> > We will strengthen the corresponding analysis and discussion in the Related Work section.
>
> **Q1.** Is the noisy-label framework introduced in Equation (1) used in prior works? If so, it would be helpful to mention and cite them.
> > We thank the reviewer for this helpful question. The formulation in Equation (1), where pseudo-labels are modeled as noisy observations with sample-dependent variance, is inspired by classical heteroscedastic regression modeling [2]. Our contribution lies in extending this well-established idea to the semi-supervised setting, where the variance is learned jointly with the pseudo-labels and plays a central role in guiding uncertainty-aware learning on unlabeled data. We will revise the manuscript to explicitly cite this related work and clarify the connection to prior literature.
>
> **M1.** Line 121: Did you mean noise level?
> > Many thanks for pointing out this unclear description. Line 121 mainly refers to the level of uncertainty or noise in the pseudo-labels varies across samples. We will provide a clearer explanation of this issue in the revised version.
>
> **M2. & M3. & M4.**
> > Manys thanks for the careful check of details. We will fix these errors in the revised version.
>
> ---
> **Reference:**
> [1] Semi-supervised deep regression with uncertainty consistency and variational model ensembling via bayesian neural networks. In  AAAI, 2023.
> [2] Pattern recognition and machine learning. In New York: springer, 2006.
> [3] Semi-supervised deep kernel learning: Regression with unlabeled data by minimizing predictive variance.  In NeurIPS, 2018.
> [4] Deep semi-supervised regression via pseudo-label filtering and calibration. In Applied Soft Computing, 2024.
> [5] Uncertainty-aware self-ensembling model for semi-supervised 3D left atrium segmentation. In MICCAI, 2019.
> [6] 3d semi-supervised learning with uncertainty-aware multi-view co-training. In CVPR, 2020.
> [7] Enhancing pseudo label quality for semi-supervised domain-generalized medical image segmentation. In AAAI, 2022.
> [8] Calibrating label distribution for class-imbalanced barely-supervised knee segmentation. In ICML, 2022.

---

> > ### Author Response · Authors · 2025-08-03
> > **Looking Forward to Receiving Feedback from Reviewer LePr**
> >
> > Hi Reviewer LePr,
> >
> > We would like to follow up to see if our response addresses your concerns or if you have any further questions. We would really appreciate the opportunity to discuss this further if our response has not already addressed your concerns. Thank you again!

---

> > > ### Comment · Reviewer_LePr · 2025-08-05
> > >
> > > Thank you for the detailed responses. The authors addressed my main concerns, and I have updated my score accordingly.
> > >
> > > I encourage the authors to incorporate a discussion on the smoothness properties of pseudo-labels, as well as relevant literature on noise-level estimation, to better contextualize their framework.
> > >
> > > Additionally, I suggest clarifying the paragraph starting at line 124. As I understand it, Equation (1), followed by the analysis in section 3.2, is introduced primarily as preliminary background for the proposed method—which is perfectly reasonable. However, I believe this intention should be stated more explicitly. Currently, the presentation may be misleading: since the noise term $\epsilon$
> > >  in Equation (1) is independent of the input $x$ it follows that for two close inputs $x_1 , x_2$, the difference between their pseudo-labels satisfies $\hat{y}_1- \hat{y}_2 \approx \epsilon_1 +\epsilon_2 \sim N(0, \sigma_1^2 + \sigma_2^2)$. Thus, under this model, proximity in input space does not necessarily imply similarity in pseudo-labels, which seems at odds with the desired smoothness property. Making this distinction clearer would improve the exposition and avoid confusion.

---

> > > > ### Author Response · Authors · 2025-08-05
> > > >
> > > > Thank you for your helpful suggestions. We will incorporate a discussion on pseudo-label smoothness and relevant noise-level estimation literature to better contextualize our method. Additionally, we will revise the paragraph around Line 124 following your valuable advice. We sincerely appreciate the time and effort you have devoted to improving our work.

---

### Official Review · Reviewer_DUVZ · 2025-06-26

**Clarity:** 3
**Significance:** 2
**Originality:** 2
**Rating:** 4
**Confidence:** 4

**Summary:**

This paper tackles the problem of learning from noisy pseudo-labels in semi-supervised regression. The authors observe that incorrect pseudo-labels with high loss can lead to overestimated uncertainty, which suppresses both noisy and difficult-but-correct samples. To address this, they introduce a bi-level optimization framework. In the inner loop, the regression model is trained using a heteroscedastic loss where the uncertainty for each pseudo-label is predicted by a lightweight auxiliary network. In the outer loop, this uncertainty learner is updated by minimizing the supervised loss on a separate batch of labeled data, effectively treating it as a hyperparameter. The goal is to adjust uncertainty estimates in a way that selectively downweights incorrect pseudo-labels without penalizing informative ones. Empirically, the method achieves consistent improvements across three semi-supervised regression benchmarks (UTKFace, IMDB-WIKI, and STS-B), particularly under low-label regimes (5% - 10%).

**Questions:**

Could strategies such as thresholding, warm-up, or moving average be incorporated to prevent the early suppression of reliable pseudo-labels? Have the authors considered or tested such techniques?

Is the bi-level structure essential, or could similar effects be achieved with dynamic reweighting or teacher-student frameworks? A justification or ablation would help validate the design choice.

How does the proposed method compare with approaches like UPS or CADR in terms of reweighting or selection mechanisms? A clearer conceptual or empirical comparison would strengthen the contribution.

If these issues are carefully addressed, I would consider raising my score.

**Ethical Concerns:**

["NO or VERY MINOR ethics concerns only"]

**Final Justification:**

The authors provided extensive clarifications and additional experiments that convincingly addressed my key concerns regarding dynamic thresholding, soft reweighting, warm-up strategies, and comparisons with closely related methods (UPS and CADR). Given these comprehensive improvements and additional insights provided during the rebuttal, I confidently raise my rating from 3 to 4, recognizing the enhanced clarity, strengthened methodology, and improved empirical validation.

**Limitations:**

yss

**Paper Formatting Concerns:**

No major formatting issues

**Quality:**

3

**Strengths And Weaknesses:**

**Strengths**
1. The paper presents a clear motivation and formulates the challenge of noisy pseudo-label suppression in semi-supervised regression with solid theoretical grounding.

2. The proposed bi-level uncertainty optimization framework is modular, lightweight, and integrates well with existing regression pipelines.

3. Experimental results show consistent improvements on three benchmarks under low-label settings, which demonstrates the general effectiveness of the method.

**Weaknesses**
1. The uncertainty learner (UL) performs worse than the base regression model in ablation studies and the explanation given is that UL cannot effectively identify hard but correct samples due to the lack of outer-loop supervision. However, this reasoning is not fully explored. The paper does not consider whether dynamic thresholding strategies such as moving thresholds or soft reweighting could reduce the risk of suppressing informative pseudo-labels during early training.

2. Following 1, while the method relies on uncertainty-aware weighting, it remains unclear whether any thresholding, warm-up, or moving average mechanisms can be used to prevent the early elimination of correct pseudo-labels. This leaves open the question of whether the bi-level design is necessary or if similar benefits could be achieved through simpler mechanisms such as dynamic reweighting or teacher-student frameworks.

3. The comparison with related methods such as UPS[1] and CADR[2] is insufficient. UPS uses Monte Carlo Dropout to estimate uncertainty and filters pseudo-labels through thresholding. CADR models sample-specific prediction errors and reweights training loss to address missing-not-at-random biases. Both approaches reflect the idea that error arises from the output but is structured by the input. Although the proposed method differs in structure by using bi-level optimization, the paper would benefit from a more detailed comparison of how different methods reweight or suppress pseudo-labels.

[1] IN DEFENSE OF PSEUDO-LABELING: AN UNCERTAINTY-AWARE PSEUDO-LABEL SELECTION FRAMEWORK FOR SEMI-SUPERVISED LEARNING
[2] ON NON-RANDOM MISSING LABELS IN SEMI-SUPERVISED LEARNING

---

> ### Author Rebuttal · Authors · 2025-07-31
>
> We appreciate very much your constructive comments on our paper. Please kindly find our response to your comments below. We hope that our response satisfactorily addresses the issues you raised. Please feel free to let us know if you have any additional concerns or questions.
>
> **W1. & Q2.** The paper does not consider whether dynamic thresholding strategies such as moving thresholds or soft reweighting could reduce the risk of suppressing informative pseudo-labels during early training.
> > Thanks for pointing this out. As recommended, we conducted additional experiments using both **moving threshold and soft reweighting strategies**, with the corresponding results are presented below.
> >  - (**Implementation Details**). **As for Moving Threshold Strategy**, we have compared with a dynamic thresholding approach SimRegMatch [4] in Table 2 of the main text (also shown below), where the threshold is updated after each training iteration and set to the 𝐾-th percentile of uncertainty values computed from the current batch of unlabeled data; **For Soft Reweighting**, we follow the method in [5],  treating per-sample uncertainty $\sigma_j^2$ as a learnable parameter to weight the regression loss; The term "Baseline" in the table refers to omitting the uncertainty learner and using a fixed uncertainty $\sigma^2_j = 1$ for all unlabeled samples, as discussed in Lines 325–327 of Section 4.3.
> >  - (**Results Analysis**). As shown in the table, the **moving threshold method underperforms compared to both the baseline and our proposed approach**. We attribute this to **its inherently hard-thresholding nature**: although it dynamically updates the threshold and uses a Top-K selection strategy, it tends to discard many potentially informative pseudo-labels. **Soft reweighting performs better than the moving threshold method but is still inferior to our approach.** Our uncertainty-aware method can actually be viewed as a form of soft reweighting, since it continuously adjusts each sample’s loss contribution based on estimated uncertainty. However, there is a fundamental difference in how the weights are generated. Soft reweighting directly learns sample-wise weights, which may lead to instability and overfitting, especially during early training, whereas our approach adaptively determines the weights, leading to more stable and robust performance.
> >
> | Method| 5%  |        | 10%       |        | 20%       |        |
> |--------------------|----------|--------|----------|--------|----------|--------|
> |                    | MAE      | $R^2$     | MAE      | $R^2$     | MAE      | $R^2$     |
> | Baseline            | 9.512 ± 0.076     | 0.651 ± 0.005 | 8.864  ±  0.083 | 0.683 ± 0.004 | 8.605 ± 0.060  | 0.696 ±  0.002 |
> | SimRegMatch [4] | 9.908 ± 0.097  | 0.628 ± 0.004 | 9.110 ± 0.166  | 0.665 ± 0.007 | 8.587 ± 0.094  | 0.693 ± 0.006|
> | Soft Reweighting [5]   | 9.475 ±  0.036  | 0.652 ± 0.001| 8.779 ± 0.029  | 0.686 ± 0.002| 8.399 ± 0.058 | 0.706 ± 0.003 |
> | Ours               | **9.177 ± 0.061**  | **0.664 ± 0.003**| **8.539 ±  0.065** | **0.695 ±  0.003**| **8.166 ± 0.071** | **0.712 ± 0.002** |
> |
>
> **W2. & Q1.** Following 1, while the method relies on uncertainty-aware weighting, it remains unclear whether any thresholding, warm-up, or moving average mechanisms can be used to prevent the early elimination of correct pseudo-labels. This leaves open the question of whether the bi-level design is necessary or if similar benefits could be achieved through simpler mechanisms such as dynamic reweighting or teacher-student frameworks.
>
> > Thank you for your thoughtful feedback. To address your concerns, we conducted additional experiments incorporating **warm-up, and moving average mechanisms**, in order to further assess the necessity of our bi-level design.
> > - (**Implementation Details**). For the warm-up strategy, following [3], **we trained the model exclusively on labeled data during the initial epochs** to mitigate premature filtering of pseudo-labels. For the moving average mechanism, we **smooth model parameters over training iterations** [6].
> > - (**Results Analysis**). From the following table, we can observe that **our proposed method significantly outperforms both the warm-up and moving average methods** across multiple settings, highlighting the effectiveness and necessity of the bi-level design.
> >
> | Method             | 5%              |        | 10%             |        | 20%             |        |
> |--------------------|----------------|--------|----------------|--------|----------------|--------|
> |                    | MAE            | $R^2$     | MAE            | $R^2$     | MAE            | $R^2$     |
> | Warm-up [3]           | 9.426 ± 0.006  | 0.656 ± 0.003 | 8.904 ± 0.017  | 0.682 ± 0.004 | 8.531 ± 0.025  | 0.700 ± 0.002 |
> | Moving Average [6]      | 10.057 ± 0.056 |0.614 ± 0.003 |9.105 ± 0.049| 0.662 ± 0.003|8.502 ± 0.029 | 0.691 ± 0.002 |
> | Ours               | **9.177 ± 0.061**  | **0.664 ± 0.003** | **8.539 ± 0.065**  | **0.695 ± 0.003** | **8.166 ± 0.071**  | **0.712 ± 0.002** |
> |
>
> **W3. & Q3.** The comparison with related methods such as UPS[1] and CADR[2] is insufficient. UPS uses Monte Carlo Dropout to estimate uncertainty and filters pseudo-labels through thresholding. CADR models sample-specific prediction errors and reweights training loss to address missing-not-at-random biases. Both approaches reflect the idea that error arises from the output but is structured by the input. Although the proposed method differs in structure by using bi-level optimization, the paper would benefit from a more detailed comparison of how different methods reweight or suppress pseudo-labels.
>
> > Thanks for your valuable comment. UPS [1] and CADR [2] were originally proposed for **semi-supervised classification tasks and are not directly applicable to regression problems**. Therefore, we did not include them in our initial comparison.
> Following your suggestion, **we have now implemented and adapted these methods to our semi-supervised regression setting**.
> >
> > - **UPS** uses Monte Carlo Dropout to estimate uncertainty via the variance of multiple forward passes, and applies a threshold to identify low-confidence predictions as negative labels, thereby enabling negative learning. In contrast, regression inherently deals with continuous outputs. and defining negative targets is nontrivial. **To fairly adapt UPS to regression, we only use Monte Carlo Dropout to estimate pseudo-label uncertainty, but without negative labeling (denoted as UPS_Reg).**
> >
> > - **CADR** was designed to address the challenge of non-random (i.e., biased) label missingness in classification tasks. It achieves this by leveraging the prior distribution of labeled samples to perform inverse probability weighting (CAP module) and  adjusting per-class selection thresholds for unlabeled samples (CAI module). **Since such class-specific confidence modeling cannot be directly applied to continuous targets, we approximate this by discretizing the regression target into age bins, treating each bin as a class.**
> >
> > Based on these adaptations, we have conducted direct comparisons with both UPS and CADR in the following table. **Our method consistently achieves better performance across various settings.**
> >   - Compared with UPS_Reg, while both methods incorporate uncertainty, our method uses uncertainty-aware soft weighting within a bi-level optimization framework, yielding more stable and accurate uncertainty estimation.
> >   - The inferior performance of CADR highlights the challenge of directly adapting classification-based methods to regression tasks.
> >   - Thanks again for this suggestion, we will include the results and discussions of these comparison baselines in the revised manuscript.
>
> | Method     | 5%               |        | 10%              |        | 20%              |        |
> |------------|-----------------|--------|------------------|--------|------------------|--------|
> |            | MAE             | $R^2$    | MAE              | $R^2$     | MAE              | $R^2$    |
> | UPS_Reg [1]    | 9.430 ± 0.087   | 0.647 ± 0.006 | 8.845 ± 0.055   | 0.676 ± 0.003 | 8.415 ± 0.002   | 0.699 ± 0.001 |
> | CADR [2]      | 10.592 ± 0.091  | 0.562 ± 0.009 | 9.531 ± 0.064   | 0.622 ± 0.007 | 8.536 ± 0.056   | 0.668 ± 0.008 |
> | Ours       | **9.177 ± 0.061**   | **0.664 ± 0.003** | **8.539 ± 0.065**   | **0.695 ± 0.003** | **8.166 ± 0.071**   | **0.712 ± 0.002** |
>
> ---
> **Reference:**
> [1] Rizve, Mamshad Nayeem, et al. In defense of pseudo-labeling: An uncertainty-aware pseudo-label selection framework for semi-supervised learning. In ICLR, 2021.
> [2] On non-random missing labels in semi-supervised learning. In ICLR, 2021.
> [3] FixMatch: Simplifying semi-supervised learning with consistency and confidence. In NeurIPS, 2020.
> [4] Deep semi-supervised regression via pseudo-label filtering and calibration. In Applied Soft Computing, 2024.
> [5] What uncertainties do we need in bayesian deep learning for computer vision? In NeurIPS, 2017.
> [6] Acceleration of Stochastic Approximation by Averaging. In SIAM J. Control Optim, 1992.

---

> > ### Comment · Reviewer_DUVZ · 2025-08-01
> >
> > Thank you very much for your extensive efforts and detailed responses. I fully understand and appreciate that conducting additional experiments required significant effort.
> > I am willing to increase my rating. I will continue to follow the discussions between you and the other reviewers. Thank you.

---

> > > ### Author Response · Authors · 2025-08-03
> > > **To Reviewer DUVZ**
> > >
> > > Thank you for your kind feedback and for acknowledging our efforts. We truly appreciate your updated rating and your ongoing engagement.

---

### Official Review · Reviewer_5vPR · 2025-07-02

**Clarity:** 3
**Significance:** 3
**Originality:** 3
**Rating:** 4
**Confidence:** 4

**Summary:**

This paper investigates how to leverage pseudo-labels for semi-supervised regression (SSR). The proposed method is an uncertainty-aware framework that dynamically adjusts the influence of pseudo-labels from a bi-level optimization perspective. Extensive experiments validate the effectiveness of the proposed approach.

**Questions:**

1. Since pseudo-labels are the predictions of the current model, is the proposed method similar to the progressive identification method in [1]?

2. How efficient is the bi-level optimization method?

3. The proposed method assigns pseudo-labels to all unlabeled samples. Could it be further improved by assigning pseudo-labels only to confident samples?

[1] X. Cheng et al. Partial-Label Regression. In AAAI 2023.

**Ethical Concerns:**

["NO or VERY MINOR ethics concerns only"]

**Final Justification:**

My main concerns about this work lie in the efficiency of the proposed method and its potential for further improvement. The additional experiments provided by the authors have addressed my doubts. Therefore, I give a positive score.

**Limitations:**

N/A.

**Paper Formatting Concerns:**

In line 180, Equation (6) is missing a parenthesis.

**Quality:**

3

**Strengths And Weaknesses:**

Strengths:
1. The paper is well-organized and easy to follow.
2. Investigating how to leverage pseudo-labels in semi-supervised regression is an interesting problem.

Weaknesses:
1. The bi-level optimization mechanism is inefficient.
2. The use of pseudo-labels has already been extensively studied in related fields, such as weakly supervised learning.
3. The experiments only consider the cases where $\gamma$ = 5%, 10%, and 20%, lacking results for larger a values.

---

> ### Author Rebuttal · Authors · 2025-07-30
>
> We sincerely thank the reviewer for providing valuable feedback. Below, we respond to each point in detail. Please let us know if any concerns remain.
>
> **W1. & Q2.** The bi-level optimization mechanism is inefficient.
> > Although traditional bi-level methods are often inefficient due to the need for computing second-order derivatives, we respectfully point out that our bi-level learning framework reduces such inefficiency through algorithmic improvements, as analyzed in **Lines 217–222 of Section 3.4**. Moreover, our algorithm shows competitive computational efficiency compared to recent SSR methods, as remarked in **Table 5 of the main text**. We detail the reasons as follows:
> > + **The efficiency is mainly from a significant simplification in computing second-order derivatives.**  Unlike traditional bi-level methods that unroll the entire network to compute second-order gradients [6,7], our method simplifies this by assuming the uncertainty-learner parameters $\phi$ are associated only to the regression head. This allows gradient unrolling solely through the regression linear layer, enabling efficient outer-loop updates of $\phi$ in Eq. (8). Since the regression head contains far fewer parameters than the full regression network, this design leads to a substantial reduction in computational cost and results in significantly improved efficiency for our algorithm.
> > + **Our approach demonstrates highly competitive efficiency compared to recent SSR methods.** To verify this, we have quantified the training cost in **Table 5 of the main text (also shown below)**, which reports the average training time per iteration and GPU memory consumption on UTKFace, measured using an NVIDIA GeForce RTX 4090. As shown, our method **incurs only ~17MB additional GPU memory and less than 9ms extra training time per iteration compared to the Baseline**.   Moreover, our method outperforms recent SSR approaches [2,4,5] in both accuracy (Tables 1–3) and efficiency, demonstrating a favorable trade-off between performance and cost.
> > + Besides, **in the test stage our method requires no extra overhead compared with the Baseline model.**
> | Method | Time (ms) | GPU (MB) |
> |-|-|-|
> | Baseline | 83.0 | 5099 |
> | SimRegMatch [5] | 548.1 | 7419 |
> | UCVME [2] | 257.2  | 10057 |
> | RankUp [4] | 108.6 | 7433  |
> | Ours| **91.6** | **5116** |
> |
>
> **W2.** The use of pseudo-labels has already been extensively studied in related fields, such as weakly supervised learning.
> > We agree that pseudo-labeling has been widely studied in related areas, such as weakly supervised learning (WSL). However, our work differs from these prior studies in several important aspects:
> >  + **Application.** Most existing pseudo-labeling approaches focus on classification tasks. In contrast, our method tackles the more challenging SSR problem, where pseudo-labeling introduces additional complexity due to the continuous output space and the lack of natural thresholds to assess their reliability. To the best of our knowledge, uncertainty-aware pseudo-labeling for regression remains underexplored.
> >  + **Methodology.** Our method propose a bi-level SSR framework that explicitly optimizes the uncertainty of pseudo-labeled samples to improve the regression model's generalization when learning from potentially incorrect pseudo-labels.  In contrast to WSL methods such as Partial-Label Regression (PLR) [1],  which directly compute the prediction loss using the pseudo-label and a candidate label set,  our approach focuses on modeling pseudo-label uncertainty and dynamically adjusts the influence of pseudo-labels through a bi-level optimization process.  Furthermore, many pseudo-labeling techniques developed for SSL classification, such as the classic FixMatch, rely on fixed confidence thresholds and therefore cannot be directly applied to SSR settings due to the absence of a natural thresholding mechanism in regression tasks.
> > + **Performance.** In our comparison experiments (Table 1-3 of the main text), we include several representative pseudo-labeling methods such as SimRegMatch [5] and UCVME [2]. Our method consistently outperforms these baselines on both image and text datasets. For example, on the IMDB_WIKI dataset (Table 2), our method outperforms UCVME method by 5.7% for MAE and 4.9\% for $R^2$.  These results confirm the effectiveness and robustness of our proposed uncertainty-based pseudo-labeling technique.
> >
> > We appreciate your suggestion and will include a more comprehensive discussion of pseudo-labeling methods, including [1], in the Related Work section of the revised manuscript.
>
> **W3.** The experiments only consider the cases where $\gamma$= 5%, 10% and 20%, lacking results for larger a values.
> > + **In semi-supervised learning, a core assumption is that the amount of labeled data is significantly smaller than that of unlabeled data.** Accordingly, our experiments focus on low-label regimes with $\gamma$= 5%, 10% and 20%, which are commonly adopted in prior SSR studies [2, 3, 4] to reflect realistic scenarios. These settings allow us to evaluate the effectiveness of pseudo-labeling and uncertainty modeling under challenging conditions where they are most needed. Following this paradigm, we align our experimental setup with the commonly adopted settings in SSR [2, 3, 4].
> > +  **In response to the concern, we have added experiments with $\gamma=$40\% on the IMDB_WIKI dataset.** The results are presented in the table below and show that our method remains effective even under a higher labeled data ratio. Moreover, the relatively smaller performance gains in this setting are expected, as the performance of SSR models naturally approaches that of the fully supervised model (the upper bound model), when a large portion of the training data is labeled.
>
> | Method | Fully-Supervised| UCVME [2] | Rankup [4] | CLSS [3] | Ours |
> |-|-|-|-|-|-|
> | MAE | 7.974 ± 0.043 | 8.123 ± 0.086 | 8.072 ± 0.030 | 8.602 ± 0.066 | **8.037 ± 0.051** |
> | $R^2$| 0.724 ± 0.002 | 0.713 ± 0.002| 0.714 ± 0.003 | 0.693 ± 0.003| **0.717 ± 0.001**|
> |
>
> **Q1.** Since pseudo-labels are the predictions of the current model, is the proposed method similar to the progressive identification method in [1]？
> >  We respectfully argue that the progressive identification method (PIM) significantly differs from ours in both problem setting and methodology as follows.
> >  + **Problem Setting.** The PIM method is designed to the Partial-Label Regression, which allows each training example to be annotated with a set of candidate labels. In contrast, in the SSR setting considered in our work, each unlabeled sample is assigned a single pseudo-label predicted by the model, resulting in a one-to-one correspondence between sample and pseudo-label.
> > + **Methodology.** While both methods adopt a form of loss weighting, the underlying mechanisms differ. PIM computes the predictive loss as a weighted sum over candidate labels, where labels with smaller losses receive higher weights via a softmax function. **In contrast, our approach employs an uncertainty learner to estimate the uncertainty of each pseudo-label, and down-weights samples with higher uncertainty during training.**
> > + **Performance.**  To further address your concern, we attempted to adapt the PIM method to the SSR setting. Since PIM cannot be directly applied to SSR problems due to the absence of candidate label sets for unlabeled data, we constructed a set of pseudo-label candidates for each unlabeled sample using Monte Carlo Dropout and applied a softmax-based weighting to compute the PIM loss. As shown in the table below, **our method outperforms the adapted PIM.**
> >
> > We will include a detailed discussion and comparison of PIM in the revised version.
>
> | Method | 5% |  | 10% | | 20% | |
> |-|-|-|-|-|-|-|
> | | MAE | $R^2$ | MAE | $R^2$| MAE | $R^2$|
> | PIM [1] | 9.345 ± 0.100 | 0.659 ± 0.005 | 8.691 ± 0.021 | 0.691 ± 0.001 | 8.441 ± 0.014 | 0.704 ± 0.001 |
> | Ours  |**9.177 ± 0.061** | **0.664 ± 0.003**  | **8.539 ± 0.065**  | **0.695 ± 0.003**  |**8.166 ± 0.071**| **0.712 ± 0.002** |
> |
>
> **Q3.** Could it be further improved by assigning pseudo-labels only to confident samples?
> > We agree that assigning pseudo-labels only to confident samples can further improve performance. **To verify this, we conducted a fine-tuning experiment using only high-confidence pseudo-labeled samples on IMDB_WIKI under $\gamma$=10%.**
> > + Specifically, we defined a threshold at the $K\%$-th percentile of the uncertainty distribution over all unlabeled samples. Samples with uncertainty above this threshold were discarded.
> > + The results in the table below shows that performance gradually improves as $K$ increases from 20 to 60, and surpasses the results reported in the paper. **This supports our hypothesis that fine-tuning on confident pseudo-labeled data can refine the model and improve generalization.**
> > + However, to ensure fair comparison with other baseline methods, we did not apply fine-tuning in the main experiments.
>
> | K=| 20 | 30 | 40 | 50 | 60 | 70 | Ours |
> |-|-|-|-|-|-|-|-|
> | MAE | 8.734 | 8.593 | 8.570 | 8.530 | **8.451** | 8.503 | 8.539 |
> | $R^2$ | 0.692 | 0.696 | 0.698 | 0.697 | **0.700** | 0.697 | 0.695 |
> ---
> **Reference:**
> [1] Partial-Label Regression. In AAAI 2023.
> [2] Semi-supervised deep regression with uncertainty consistency and variational model ensembling via bayesian neural networks. In AAAI, 2023.
> [3] Semi-supervised contrastive learning for deep regression with ordinal rankings from spectral seriation. In NeurIPS, 2023.
> [4] Rankup: Boosting semi-supervised regression with an auxiliary ranking classifier. In NeurIPS, 2024.
> [5] Deep semi-supervised regression via pseudo-label filtering and calibration. In Applied Soft Computing, 2024.
> [6] Model-agnostic meta-learning for fast adaptation of deep networks. In ICML, 2017.
> [7] Learning to reweight examples for robust deep learning. In ICML, 2018.

---

> > ### Comment · Reviewer_5vPR · 2025-08-03
> >
> > Thanks for the author's rebuttal. I'm glad to see the performance improvement achieved by assigning pseudo-labels only to trustworthy samples. I will maintain my positive score.

---

> > > ### Author Response · Authors · 2025-08-03
> > > **To Reviewer 5vPR**
> > >
> > > Thank you for your feedback. We're pleased to hear that the experiment provided helpful insights, and we sincerely appreciate your kind recognition and constructive comments.

---

### Official Review · Reviewer_c7Z8 · 2025-07-02

**Clarity:** 3
**Significance:** 3
**Originality:** 3
**Rating:** 5
**Confidence:** 4

**Summary:**

This paper is on semi-supervised regression (SSR), a scenario where labeled data is limited but unlabeled data is abundant. While pseudo-labeling is widely adopted in semi-supervised classification, it is less explored for regression tasks. The main difficulty stems from the continuous nature of regression targets. Unlike classification, where pseudo-labels are discrete and confidence is easier to quantify, regression must contend with heteroscedastic noise, which makes pseudo-label reliability much harder to assess.

The authors propose an uncertainty-aware pseudo-labeling framework for SSR. The central idea is to dynamically adjust the influence of pseudo-labeled samples based on their estimated uncertainty. The method employs a bi-level optimization approach. In the inner loop, the regression model is trained on both labeled and pseudo-labeled data, where each pseudo-label is weighted according to its predicted uncertainty. In the outer loop, the parameters that determine the uncertainty are optimized to ensure better generalization on the labeled data.

A lightweight neural network, referred to as the uncertainty-learner, is introduced to estimate the uncertainty for each pseudo-labeled example. This network takes as input both the model's prediction and the pseudo-label for a given sample and produces a log-variance value representing the estimated uncertainty. This enables the model to down-weight unreliable pseudo-labels during training, thereby reducing the risk of error accumulation and overfitting to noisy signals.

The paper provides a superficial theoretical justification for this approach: The bi-level optimization implicitly aligns the learning gradients from labeled and pseudo-labeled data. MOreover, when the uncertainty estimation is accurate, the model's learning direction is guided toward solutions that generalize well, rather than merely fitting the pseudo-labeled data. This contributes to greater stability and robustness in training.

Comprehensive experiments are conducted on three benchmark datasets: UTKFace and IMDB-WIKI for age estimation from images, and STS-B for semantic similarity regression on text pairs. The method is evaluated under different proportions of labeled data, specifically at 5%, 10%, and 20%. Across all settings, the proposed approach consistently outperforms several strong baselines, including both consistency-based and uncertainty-based SSR methods. The gains are especially pronounced when labeled data is scarce.

The ablation studies further validate the design choices. Removing the uncertainty-learner or not using bi-level optimization leads to a noticeable drop in performance. The experiments also show that the uncertainty estimates produced by the model are well correlated with the actual prediction errors. Samples with higher prediction errors are assigned higher uncertainty, and their influence on model updates is reduced.Regarding efficiency, the proposed method introduces only minimal computational and memory overhead compared to standard baselines. This makes it practical for real-world applications.

In concusion, the paper presents a principled and effective solution to semi-supervised regression with noisy pseudo-labels. By modeling heteroscedastic uncertainty and optimizing it in a bi-level manner, the approach significantly enhances robustness and accuracy, particularly in data-scarce regimes. The results suggest that careful handling of uncertainty is crucial for the success of SSR, and the framework established here sets a new standard for future research in the field.

**Questions:**

see above

**Ethical Concerns:**

["NO or VERY MINOR ethics concerns only"]

**Final Justification:**

See summary of strenghts and weaknesses

**Limitations:**

see above

**Quality:**

3

**Strengths And Weaknesses:**

I think this is a prime example of a borderline paper. The methodology, in particular the bi-level optimization for learning uncertainty in pseudo-labeling for heteroscedastic SSR, is novel and the experiments speak for themselves. However, the theoretical analysis is pretty slim and not that insightful. Theorem 1 states a mere re-write of the optimizaiton problem. Directly seeing the implicit regularization of the gradients in eq.9 is nice, but also already apparent to the educated reader by eq. 5 and 6. I think the paper would greatly benefit from a deeper theoretical underpinning. (That being said, I want to re-iterate that I really like the methodological ideas and find the experimental results very convincing.) Here is an actionable suggestion of how to add some theoretical motivation: The authors could easily prove (approximate) Bayes-optimality of \theta^* and \phi^* by framing the bi-level optimization problem as decision problem with mixed decision rules. Building on related work on uncertainty-aware pseudo-label selection [1-4], in particular [2] which embeds pseudo-label selection in decision theory, it is easy to see that the calibration via $\sigma^2$ corresponds to a probability weight vector of all actions (=pseudo-labeled data). Let me maybe add some further explanations: the decision-theoretic embedding and Bayes-optimality proves in [2] are concerned with pseudo-label *selection", where a only a subset of pseudo-labeled data is added to labeled data iteratively ("self-training"). I am aware the setup is different here. However, the decision-theoretic embedding still holds by considering mixed-action decision theory, also known as randomized decision theory, which involves making decisions by using a weighting vector (probability distribution) over multiple actions, rather than choosing a single action. The calibration via the $\sigma^2$ corresponds to exactly such a weighting over all actions (=pseudo-labels). By framing as an adequate decision problem, it can be seen that the proposed method corresponds to the best strategy given all available information and uncertainties (i.e., Bayes action).

[1] Rizve, Mamshad Nayeem, et al. "In Defense of Pseudo-Labeling: An Uncertainty-Aware Pseudo-label Selection Framework for Semi-Supervised Learning." International Conference on Learning Representations. 2021.

[2] Rodemann, Julian, et al. "Approximately Bayes-optimal pseudo-label selection." Uncertainty in Artificial Intelligence. 2023.

[3] Rodemann, Julian, et al. "In all likelihoods: Robust selection of pseudo-labeled data." International Symposium on Imprecise Probability: Theories and Applications. 2023.

[4] Dietrich, Stefan, et al. "Semi-supervised learning guided by the generalized Bayes rule under soft revision." International Conference on Soft Methods in Probability and Statistics. 2024

Minor remarks:

- Index in second summand of eq.3 should be j rather than x_j^u, as the latter is not used in the sum. Please also update the index set accordingly to $\|\mathcal B_u \|$
- redundant use of "optimization" in proof of theorem 1: "irrelevant to optimization of $\Phi_t$, which can beomitted in optimization"

---

> ### Author Rebuttal · Authors · 2025-07-30
>
> We sincerely thank you for the encouraging feedback on the novelty of our bi-level optimization framework for learning uncertainty and the strength of our experimental results, as well as the constructive suggestions for strengthening the theoretical aspect from a Bayesian decision perspective. Below, we respond to each point in detail. Please let us know if any concerns remain.
>
> **W1.**  The theoretical analysis is pretty slim and not that insightful. Theorem 1 states a mere re-write of the optimization problem.
>
> > We agree that Theorem 1 may appear to be a direct derivation from the optimization procedure.  However, we respectfully point out that this  theoretical result plays an important role in clarifying the mechanism behind our method, and contributes to the broader understanding of uncertainty-based learning in two respects:
> >
> > + **Theoretical insight into the mechanism.** This theorem provides a formal explanation of how uncertainty-aware weighting affects the training process.  In particular, it shows that our method automatically focuses on pseudo-labeled samples whose gradients are similar to those from labeled data, and reduces the impact of samples with higher predictive uncertainty. This insight helps explain why our method remains robust under incorrect or uncertain supervision. By selectively down-weighting the influence of noisy gradients,  it promotes more stable training and better generalization. In this way,   we think the theorem not only reflects the optimization procedure, but also helps explain more clearly why the method works in practice.
> > + **Theoretical bridge to gradient matching.** The theorem bridges our bi-level optimization framework with the emerging principle of gradient matching, a concept that has proven effective in tasks such as domain adaptation [5], dataset distillation [6], and few-shot learning [7]. This connection not only grounds our approach in a broader theoretical context but also suggests its potential applicability to other settings.   By showing that our uncertainty-based weighting tends to select pseudo-labels whose gradients align with those from labeled data, the theorem places our method within a broader family of alignment-based strategies.  This underscores its potential to serve as a foundation for semi-supervised learning methods that leverage gradient matching.
>
> **Q1.** Here is an actionable suggestion of how to add some theoretical motivation: The authors could easily prove (approximate) Bayes-optimality of $ \theta^* $ and $ \phi^* $ by framing the bi-level optimization problem as decision problem with mixed decision rules.
>
> > Following your valuable suggestion, we made an effort to connect our method to Bayesian decision theory to further interpret our bi-level framework.
> >
> > + To make the theoretical analysis more tractable, we restate our bi-level learning formulation with a slight simplification. Specifically, we let the predictive variance $\sigma_j^2 = g_\phi(x_j)$, rather than using its logarithm as in the main text. This modification only involves a log-transform and does not alter the essential behavior of our model. The **inner-loop optimization** becomes: $\theta^{\*}(\phi) = \arg\min_\theta \sum_{i=1}^n (y_i - f_\theta(x_i))^2 + \lambda \sum_{j=1}^m \left[ \frac{1}{g_\phi(x_j)} (\hat y_j - f_\theta(x_j))^2 + \log g_\phi(x_j) \right],$ and the **outer-loop optimization** is: $\phi^* = \arg\min_\phi \sum_{(x_k, y_k) \in \hat{\mathcal B}\_l} \left( y_k - f_{\theta^{\*}(\phi)} (x_k) \right)^2,$ which seeks optimal uncertainty weights $g_\phi(x_j)$ to guide the model $f_{\theta^*(\phi)}$ toward better generalization.
> >
> > + To link our bi-level learning framework to a Bayesian decision problem, we formulate the following decision problem. Concretely, the uncertainty estimator $ g_\phi: \mathcal{X} \to \mathbb{R}^+ $ defines a **randomized (mixed) decision rule** by assigning weights to unlabeled samples $ x_j \in \mathcal D_u $. The model parameter $ \theta $ is treated as the latent state, and we define the utility function as: $u(g_\phi, \theta) = \log P(\mathcal D_l\cup D_u|\theta)=\log \prod_{x_i\in\mathcal D_l} \mathcal N (y_i \mid f_\theta(x_i), \sigma^2) \prod_{x_j\in\mathcal D_u}\mathcal N (\hat y_j \mid f_\theta(x_j), g_\phi(x_j)), $ where $ \hat y_j $ denotes the pseudo-label for $ x_j $. Under this formulation, the **Bayes-optimal uncertainty function** $ g_\phi^* $ is the one that maximizes the expected utility with respect to the posterior distribution over $\theta$, i.e., $\phi^* = \arg\max_{\phi} \mathbb E_{\theta \mid \mathcal D_l}[u(g_\phi, \theta)].$
> > + We now show an intuition connection between our bi-level optimization and this Bayesian formulation. Specifically, the inner-loop optimization is equivalent to minimizing the negative log-likelihood of the data under the generative model  defined earlier. Therefore, $\theta^\*(\phi)$ can be interpreted as a MAP estimate of model parameters $\theta$, conditioned on the fixed uncertainty function $g_\phi$. The outer-loop optimization aims to find the optimal uncertainty weights that lead to a model $f_{\theta^\*(\phi)}$ with best generalization performance. While this is not a direct maximization of expected utility $\mathbb E_{\theta}[u(g_\phi, \theta)]$, we argue the following: (i) If the posterior $p(\theta \mid \mathcal{D}_l)$ is sharply peaked (posterior concentration), and (ii) if the outer-loop loss over $\hat{\mathcal{B}}_l$ is a reliable proxy for risk under $p(x, y)$, then minimizing the outer-loop loss approximately maximizes the expected utility, and thus $\phi^*$ approximates the Bayes-optimal mixed-action policy.
> >
> >  Although we were unable to provide a full theoretical equivalence between our optimization procedure and the Bayes-optimal policy under the mixed-action decision framework due to the limited time available during the rebuttal phase,  we believe our preliminary formulation offers a promising starting point. In particular, the Bayesian framework in [2] on learning under noisy supervision may offer useful insights for further formalizing our uncertainty modeling.
> >
> > We sincerely thank the reviewer again for this valuable suggestion. It opens a new door for our future research to rigorously bridge Bayesian decision theory and optimization-driven uncertainty learning, as well as to address confirmation bias in semi-supervised regression from a principled Bayesian perspective.
>
> **M1.** Index in second summand of eq.3 should be $j$ rather than $x_j^u$.
> > Thank you for pointing out this minor but important detail! We will fix it in the revised version.
>
> **M2.** Redundant use of "optimization" in proof of theorem 1: "irrelevant to optimization of $\Phi_t$, which can beomitted in optimization".
> > Thank you for this helpful remark! We will remove the redundant use of "optimization" in the proof of Theorem 1 and improve the wording accordingly in the revised version.
>
> ---
> > **Reference:**
> [1] In Defense of Pseudo-Labeling: An Uncertainty-Aware Pseudo-label Selection Framework for Semi-Supervised Learning. In ICLR, 2021.
> [2] Approximately Bayes-optimal pseudo-label selection. In UAI, 2023.
> [3] In all likelihoods: Robust selection of pseudo-labeled data. In ISIPTA, 2023.
> [4] Semi-supervised learning guided by the generalized Bayes rule under soft revision. In ICSMPS, 2024.
> [5] Gradient Matching for Domain Generalization. In ICLR, 2022.
> [6] Dataset Condensation with Gradient Matching. In ICLR, 2021.
> [7] Gradient matching generative networks for zero-shot learning. In CVPR, 2019.

---

> > ### Author Response · Authors · 2025-08-03
> > **Looking Forward to Receiving Feedback from Reviewer c7Z8**
> >
> > Hi Reviewer c7Z8,
> >
> > Thanks for your constructive comments. We would like to follow up to see if our response addresses your concerns or if you have any further questions.  Thanks for your attention and best regards.

---

> > > ### Comment · Reviewer_c7Z8 · 2025-08-04
> > >
> > > Dear Authors,
> > >
> > > Thank you for your thoughtful response. The decision-theoretic embedding based on [2,3] looks promising and adds some theoretical depth to the otherwise already quite substantial contribution. In light of these promised changes to the manuscript, I am ready to raise my scores.
> > >
> > > Best,
> > > A reviewer

---

> > > > ### Author Response · Authors · 2025-08-05
> > > > **To Reviewer c7Z8**
> > > >
> > > > Thank you for your feedback. We're pleased to hear that our rebuttal addresses your concerns, and we sincerely appreciate your positive assessment and support for raising the score.

---

### Note · Authors · 2025-08-12

We sincerely thank all reviewers for their valuable comments, and we greatly appreciate the AC's dedicated efforts throughout the review process. We are particularly grateful to all reviewers for the opportunity to address their concerns and for their positive scores to our work. The main points are summarized as follows:
+ For Reviewer c7Z8, we elaborate on the insights and significance of Theorem 1, and further link our method to Bayesian decision theory.  We are pleased that the reviewer recognized that it adds theoretical depth to a substantial contribution and expressed readiness to raise the score,  for which we are truly grateful.
+ For Reviewer 5vPR, we present a detailed training cost analysis to show the efficiency of our bi-level algorithm and clearly differentiate our pseudo-labeling approach from those employed in weakly supervised learning.  Additionally, we conduct experiments with $\gamma$=40\% and perform fine-tuning using only high-confidence pseudo-labeled samples.  We are pleased that the reviewer appreciated the performance improvement achieved by assigning pseudo-labels to trustworthy samples and decided to maintain the positive score, which we greatly appreciate.
 + For Reviewer DUVZ,  we incorporate additonal comparison experiments covering the approaches suggested by the reviewer, including dynamic thresholding, soft reweighting, warm-up, and moving average methods. The results show that the bi-level framework is crucial for enabling our method to achieve superior performance in semi-supervised regression. We greatly appreciate the reviewer's acknowledgment of the significant effort in conducting these additional experiments and the willingness to increase the rating.
+ For Reviewer LePr, we rigorously verify the smoothness property both theoretically and empirically, and present preliminary results on the effect of our method on generalization. Additionally, we provide an expanded discussion of noise-level estimation techniques.  We appreciate that the reviewer acknowledges our efforts in addressing the main concerns, and updated the score accordingly.

Based on the reviewers' feedback, we will incorporate their constructive suggestions to further improve both the theoretical and empirical aspects of our paper. We are pleased that all reviewers are satisfied with our responses and have provided positive evaluations. We sincerely thank the reviewers for recognizing the novelty and potential impact of our work.

---

### Decision · Program_Chairs · 2025-09-17

**Decision:**

Accept (poster)

**Comment:**

The paper introduces an uncertainty-aware pseudo-labeling framework for semi-supervised regression. It tackles the challenge of unreliable continuous pseudo-labels under heteroscedastic noise. Using a bi-level optimization strategy, it dynamically adjusts pseudo-label influence by jointly minimizing empirical risk and optimizing uncertainty estimates for better generalization. After the rebuttal and discussions between the reviewers and authors, the paper received the final scores of A, BA, BA, BA. The reviewers largely agreed that the paper makes a substantial and novel methodological contribution, with strong empirical validation and useful theoretical work. The strengths include a clear motivation, robust bi-level pseudo-labeling design, and strong results on several benchmarks. The remaining weaknesses are that the theoretical analysis is still limited, offering only preliminary insights rather than a deep characterization of when and why the method works best; the noise modeling assumption may not fully capture real pseudo-label generation; and some reviewers felt the role and design of the uncertainty learner could have been discussed in more depth. However, these weaknesses were not considered significant.